# Composite Flow Matching for Reinforcement Learning with Shifted-Dynamics Data

**Lingkai Kong**\*   **Haichuan Wang**\*   **Tonghan Wang**\*   **Guojun Xiong**   **Milind Tambe**
School of Engineering and Applied Sciences
Harvard University

## Abstract

Incorporating pre-collected offline data can substantially improve the sample efficiency of reinforcement learning (RL), but its benefits can break down when the transition dynamics in the offline dataset differ from those encountered online. Existing approaches typically mitigate this issue by penalizing or filtering offline transitions in regions with large dynamics gap. However, their dynamics-gap estimators often rely on KL divergence or mutual information, which can be ill-defined when offline and online dynamics have mismatched support. To address this challenge, we propose COMPFLOW, a principled framework built on the theoretical connection between flow matching and optimal transport. Specifically, we model the online dynamics as a conditional flow built upon the output distribution of a pretrained offline flow, rather than learning it directly from a Gaussian prior. This composite structure provides two advantages: (1) improved generalization when learning online dynamics under limited interaction data, and (2) a well-defined and stable estimate of the dynamics gap via the Wasserstein distance between offline and online transitions. Building on this dynamics-gap estimator, we further develop an optimistic active data collection strategy that prioritizes exploration in high-gap regions, and show theoretically that it reduces the performance gap to the optimal policy. Empirically, COMPFLOW consistently outperforms strong baselines across a range of RL benchmarks with shifted-dynamics data.

## 1 Introduction

Reinforcement Learning (RL) has demonstrated remarkable performance in complex sequential decision-making tasks such as playing Go and Atari games [53, 55], supported by access to large amounts of online interactions with the environment. However, in many real-world domains such as robotics [26, 60], healthcare [69], and wildlife conservation [29, 30, 66, 67], access to such interactions is often prohibitively expensive, unsafe, or infeasible. The limited availability of interactions presents a major challenge for learning effective and reliable policies. To address this challenge and improve sample efficiency during online training, a promising strategy is to incorporate a pre-collected offline dataset generated by a previous policy [44, 59]. This approach enables the agent to learn from a broader set of experiences, which can help accelerate learning and improve performance [59].

A critical challenge in online RL with offline data arises when the transition dynamics in the offline dataset differ from those in the online environment where the agent actively interacts and learns [49]. This issue, commonly referred to as *shifted dynamics*, can create severe distribution mismatch, bias policy updates, destabilize learning, and ultimately degrade performance. For instance, in robotics, the transition dynamics specify how the robot's state (e.g., position and velocity) evolves after executing an action. During deployment, changes in physical parameters such as surface friction can alter this state evolution, causing the true next-state distribution to deviate from that implied by the historical data. Similarly, in conservation planning, the transition dynamics describe how the spatial risk of

---

\*Equal contribution. Corresponding author: `lingkaikong@g.harvard.edu`

39th Conference on Neural Information Processing Systems (NeurIPS 2025).

poaching evolves in response to patrol actions. Data collected in one region may not transfer to another because differences in terrain, accessibility, and human activity patterns lead to different state transitions and behavioral responses under the same patrol strategy [66].

To address these challenges, existing methods either penalize the rewards or value estimates of offline transitions with high dynamics gap [37, 46], or filter out such transitions entirely [64]. However, these approaches face key limitations. Most notably, the estimation of the dynamics gap typically relies on KL divergence or mutual information, both of which can be ill-defined when the offline and online transition dynamics have different supports [2, 48].

In this paper, we propose COMPFLOW, a new method for RL with shifted-dynamics data that leverages the theoretical connection between flow matching and optimal transport. We model the transition dynamics of the online environment using a composite flow architecture, where the online flow is defined on top of the output distribution of a learned offline flow rather than being initialized from a Gaussian prior. This design enables a principled estimation of the dynamics gap using the Wasserstein distance between the offline and online transition dynamics. On the theoretical side, we show that the com-

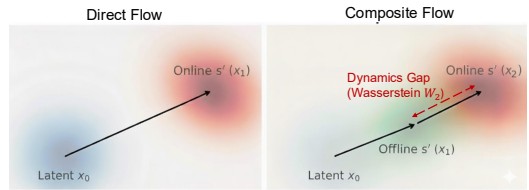

Figure 1: Comparison between direct and composite flow matching. Composite flow first transports from a Gaussian latent variable to the offline transition distribution, then adapts to the online distribution via optimal transport flow matching.

posite flow formulation reduces generalization error compared standard flow matching by reusing structural knowledge embedded in the offline data, particularly when online interactions are limited.

Building on the dynamics gap estimated by composite flow matching using the Wasserstein distance, we go beyond selectively merging offline transitions with low dynamics gap and further propose an active data collection strategy that targets regions in the online environment where the dynamics gap relative to the offline data is high. Such regions are often underrepresented in the replay buffer due to the dominance of low-gap samples. Our theoretical analysis shows that targeted exploration in these high-gap regions can further close the performance gap with respect to the optimal policy.

Our contributions are summarized as follows: (1) We introduce a composite flow model that estimates the dynamics gap by computing the Wasserstein distance between conditional transition distributions. We provide theoretical analysis showing that this approach achieves lower generalization error compared to learning the online dynamics from scratch. (2) Leveraging this principled estimation, we propose a new data collection strategy that encourages the policy to actively explore regions with high dynamics gap. We also provide a theoretical analysis of its performance benefits. (3) We empirically validate our method on various RL benchmarks with shifted dynamics and demonstrate that COMPFLOW outperforms or matches state-of-the-art baselines across these tasks.

## 2   Problem Statement and Background

### 2.1   Problem Definition

We consider two infinite-horizon MDPs: $\mathcal{M}_{\text{off}} := (\mathcal{S}, \mathcal{A}, p_{\text{off}}, r, \gamma)$ and $\mathcal{M}_{\text{on}} := (\mathcal{S}, \mathcal{A}, p_{\text{on}}, r, \gamma)$, sharing the same state/action spaces, reward function $r : \mathcal{S} \times \mathcal{A} \to \mathbb{R}$, and discount factor $\gamma \in (0, 1)$, but differing in transition dynamics:

$$p_{\text{off}}(s'|s, a) \neq p_{\text{on}}(s'|s, a) \quad \text{for some } (s, a).$$

We assume rewards are bounded, i.e., $|r(s, a)| \leq r_{\max}$ for all $s, a$. For any policy $\pi$, let $(s_t, a_t)_{t \geq 0}$ be the trajectory generated by $\mathcal{M}$ and $\pi$. We define the discounted state–action visitation (occupancy) measure as $\rho_{\mathcal{M}}^{\pi}(s, a) := (1 - \gamma)\, \mathbb{E}[\sum_{t=0}^{\infty} \gamma^t \, \mathbf{1}\{s_t = s,\, a_t = a\}]$, and the discounted state visitation as $d_{\mathcal{M}}^{\pi}(s) := \sum_a \rho_{\mathcal{M}}^{\pi}(s, a)$. The expected return is $\eta_{\mathcal{M}}(\pi) := \mathbb{E}_{(s,a) \sim \rho_{\mathcal{M}}^{\pi}}[r(s, a)]$.

**Definition 2.1** (Online Policy Learning with Shifted-Dynamics Offline Data). Given an offline dataset $\mathcal{D}_{\text{off}} = \{(s_i, a_i, s_i', r_i)\}_{i=1}^N$ from $\mathcal{M}_{\text{off}}$ and limited online access to $\mathcal{M}_{\text{on}}$, the objective is to learn a policy $\pi$ maximizing $\eta_{\mathcal{M}_{\text{on}}}(\pi)$, ideally approaching $\eta_{\mathcal{M}_{\text{on}}}(\pi^\star)$, where $\pi^\star := \arg\max_\pi \eta_{\mathcal{M}_{\text{on}}}(\pi)$.

**Lemma 2.2** (Return Bound between Two Environments [37]). *Let the empirical behavior policy in* $\mathcal{D}_{\mathrm{off}}$ *be* $\pi_{\mathcal{D}_{\mathrm{off}}}(a \mid s)$. *Define* $C_1 = \frac{2r_{\max}}{(1-\gamma)^2}$. *Then for any policy* $\pi$,

$$
\eta_{\mathcal{M}_{\mathrm{on}}}(\pi) - \eta_{\mathcal{M}_{\mathrm{off}}}(\pi) \geq -2C_1 \, \mathbb{E}_{(s,a) \sim \rho_{\mathcal{M}}^{\pi_{\mathcal{D}_{\mathrm{off}}}}, s' \sim p_{\mathrm{off}}} \left[ D_{\mathrm{TV}}(\pi(\cdot|s') \parallel \pi_{\mathcal{D}_{\mathrm{off}}}(\cdot|s')) \right]
$$
$$
- C_1 \, \mathbb{E}_{(s,a) \sim \rho_{\mathcal{M}}^{\pi_{\mathcal{D}_{\mathrm{off}}}}} \left[ D_{\mathrm{TV}}(p_{\mathrm{on}}(\cdot|s,a) \parallel p_{\mathrm{off}}(\cdot|s,a)) \right].
$$

This bound highlights two key sources of return gap between domains: (1) the mismatch between the learned policy and the behavior policy in the offline dataset, and (2) the shift in environment dynamics. The former can be mitigated through behavior cloning [37, 65], while the latter can be addressed by filtering out source transitions with large dynamics gaps [39, 64]. A central challenge lies in accurately estimating this gap. Existing methods often rely on KL divergence or mutual information [37, 46], which can be ill-defined when the two dynamics have different supports.

## 2.2 Flow Matching

In this paper, we will adopt flow matching [1, 34, 36] to model the transition dynamics, owing to its ability to capture complex distributions. Flow Matching (FM) offers a simpler alternative to denoising diffusion models [21, 57], which are typically formulated using stochastic differential equations (SDEs). In contrast, FM is based on deterministic ordinary differential equations (ODEs), providing advantages such faster inference, and often improved sample quality.

The goal of FM is to learn a time-dependent velocity field $v_\theta(x, t) : \mathbb{R}^d \times [0, 1] \to \mathbb{R}^d$, parameterized by $\theta$, which defines a flow map $\psi_\theta(x_0, t)$. This map is the solution to the ODE

$$
\frac{d}{dt}\psi_\theta(x_0, t) = v_\theta(\psi_\theta(x_0, t), t), \quad \psi_\theta(x_0, 0) = x_0,
$$

and transports samples from a simple source distribution $p_0(x)$ (e.g., an isotropic Gaussian) at time $t = 0$ to a target distribution $p_1(x)$ at time $t = 1$. In practice, generating a sample from the target distribution involves drawing $x_0 \sim p_0(x)$ and integrating the learned ODE to obtain $x_1 = \psi_\theta(x_0, 1)$.

A commonly used training objective in flow matching is the *linear path matching loss*

$$
\mathcal{L}(\theta) = \mathbb{E}_{t \sim \mathcal{U}[0,1], \, x_0 \sim p_0(x_0), x_1 \sim p_1(x_1)} \left[ \| v_\theta(x_t, t) - (x_1 - x_0) \|_2^2 \right], \tag{1}
$$

where $x_t = (1 - t)x_0 + tx_1$ denotes the linear interpolation between $x_0$ and $x_1$. Using linear interpolation paths encourages the learned flow to follow nearly straight-line trajectories, which reduces discretization error and improves the computational efficiency of ODE solvers during sampling [36].

## 2.3 Optimal Transport Flow Matching and Wasserstein Distance

Optimal Transport Flow Matching (OT-FM) establishes a direct connection between flow-based modeling and Optimal Transport (OT), providing a principled framework for quantifying the discrepancy between two distributions. This connection is especially valuable in our setting, where a key challenge is to measure the distance between two conditional transition distributions.

Let $c : \mathbb{R}^d \times \mathbb{R}^d \to \mathbb{R}$ be a cost function. Optimal transport aims to find a coupling $q^\star \in \Pi(p_0, p_1)$—a joint distribution with marginals $p_0$ and $p_1$—that minimizes the expected transport cost:

$$
\inf_{q \in \Pi(p_0, p_1)} \int c(x_0, x_1) \, \mathrm{d}q(x_0, x_1).
$$

This minimum defines the Wasserstein distance $W_c(p_0, p_1)$ between the two distributions under the cost function $c$. When $c(x_0, x_1) = \|x_0 - x_1\|^2$, the resulting distance is known as the squared 2-Wasserstein distance.

In the training objective of Eq. 1, when sample pairs $(x_0, x_1)$ are drawn from the optimal coupling $q^\star$, the flow model trained with the linear path matching loss learns a vector field that approximates the optimal transport plan. The transport cost of the learned flow approximates the Wasserstein distance

$$
\mathbb{E}_{x_0 \sim p_0} \left[ \| \psi_\theta(x_0, 1) - x_0 \|_2^2 \right] \approx W_2^2(p_0, p_1).
$$

For theoretical justification, see Theorem 4.2 in [47].

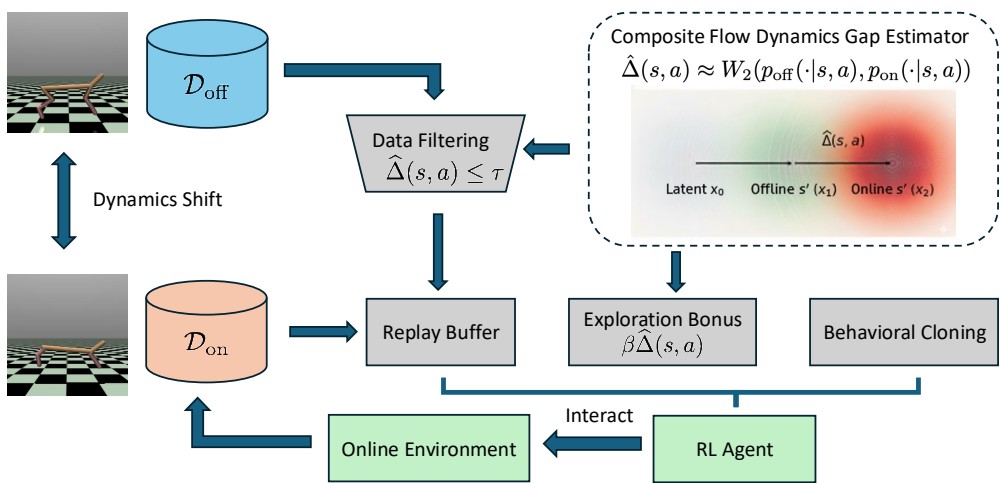

Figure 2: Overall Framework of COMPFLOW. To estimate the dynamics gap, we propose composite flow matching, which computes the Wasserstein distance between offline and online transition dynamics. Guided by the estimated dynamics gap, we augment policy training with offline transitions that exhibit low discrepancy from the online dynamics, and incorporate a behavior cloning objective to stabilize learning. To enhance data diversity and facilitate adaptation, we further encourage exploration in regions with high dynamics gap.

In practice, the optimal coupling $\pi^\star$ is approximated using mini-batches by solving a discrete OT problem between empirical samples from $p_0$ and $p_1$. The use of mini-batch OT has also been shown to implicitly regularize the transport plan [13, 14], as the stochasticity from independently sampled batches induces behavior similar to entropic regularization [9]. Further details of the OT-FM training procedure are provided in Appendix C.

## 3 Proposed Method

In this section, we introduce our method, COMPFLOW. We begin by presenting a composite flow matching approach to estimate the dynamics gap via Wasserstein distance. Next, we propose a data collection strategy that actively explores high dynamics-gap regions and provide a theoretical analysis of its benefits. Finally, we describe the practical implementation details of our method.

### 3.1 Estimating Dynamics Gap via Composite Flow

To estimate the gap between dynamics, we first learn the transition models for both the offline dataset and the online environment. Flow matching provides a flexible framework for modeling complex transition dynamics; however, the limited number of samples available in the online environment presents a significant challenge. Training separate flow models for each environment can result in poor generalization in the online environment, as the model may overfit to the small amount of available data.

To mitigate this, we propose a **composite flow** formulation. Instead of learning the online transition model $p_{\text{on}}(s'|s, a)$ from scratch, we leverage structural knowledge from a well-trained offline model $p_{\text{off}}(s'|s, a)$. This enables to incorporate prior knowledge to improve the generalization.

**Offline flow.** We begin by learning a conditional flow model for the offline data. Let $x_0 \sim \mathcal{N}(0, \mathbf{I})$ be the initial latent variable. The offline flow map $\psi_\theta^{\text{off}}(x_0, t|s, a)$ is defined as the solution to the following ODE:

$$\frac{\mathrm{d}}{\mathrm{d}t}\psi_\theta^{\text{off}}(x_0, t|s, a) = v_\theta^{\text{off}}(\psi_\theta^{\text{off}}(x_0, t|s, a), t, s, a), \quad \psi_\theta^{\text{off}}(x_0, 0|s, a) = x_0.$$

Solving this ODE from $t = 0$ to $1$ produces an intermediate representation: $x_1 = \psi_\theta^{\text{off}}(x_0, 1|s, a)$.

**Online flow.** Instead of learning a online flow directly from a Gaussian prior, we initialize it from the intermediate representation $x_1$ produced by the offline flow. The target flow map $\psi_\phi^{\text{on}}(x, t|s, a)$ is defined as:

$$\frac{\mathrm{d}}{\mathrm{d}t}\psi_\phi^{\text{on}}(x, t|s, a) = v_\phi^{\text{on}}(\psi_\phi^{\text{on}}(x, t|s, a), t, s, a), \quad \psi_\phi^{\text{on}}(x, 1|s, a) = x_1.$$

Solving this ODE from $t = 1$ to $2$ yields the final prediction for the online environment: $s' \doteq x_2 = \psi_\phi^{\mathrm{on}}(x_1, 2|s, a)$.

We now show that when the offline flow induces a distribution $\hat{p}_{\mathrm{off}}(s'|s, a)$ that is closer to $p_{\mathrm{on}}(s'|s, a)$ than the standard Gaussian distribution, the proposed composite flow method enjoys a smaller generalization error bound.

**Theorem 3.1** (Conditions for Composite Flow Yielding Smaller Errors). *Assume that composite flow and direct flow share the same hypothesis class $\mathcal{H}$ of (measurable) vector fields $v : \mathbb{R}^d \times [0,1] \to \mathbb{R}^d$, and that there exists $B \in (0, \infty)$ such that $\sup_{v \in \mathcal{H}} \sup_{x \in \mathbb{R}^d, t \in [0,1]} \|v(x, t)\|_2 \le B$. Also, assume that $\max\{\mathrm{Tr}(\Sigma_{\mathrm{on}}), \mathrm{Tr}(\widehat{\Sigma}_{\mathrm{off}})\} \le C_{\mathrm{TR}}$ for some $C_{\mathrm{TR}}$. The composite flow enjoys a strictly tighter high-probability generalization bound than the direct flow if and only if*

$$W_2(p_G, p_{\mathrm{on}}) > W_2(\hat{p}_{\mathrm{off}}, p_{\mathrm{on}}) \tag{2}$$

*Here, $p_G := \mathcal{N}(0, I_d)$, $\Sigma_{\mathrm{on}}$ is the covariance of $p_{\mathrm{on}}$, and $\widehat{\Sigma}_{\mathrm{off}}$ is the covariance of $\hat{p}_{\mathrm{off}}$.*

*Remark* 3.2. In our setting, we assume that the offline data and the online environment share meaningful similarities. Given the abundance of offline data available to accurately learn the offline flow, this assumption is expected to hold well.

To compute the Wasserstein distance between the offline and online transition dynamics, we can use the OT-FM objective to train the online flow, initialized from the offline flow distribution. However, when the state-action pair $(s, a)$ lies in a continuous space, it is not feasible to obtain a batch of samples corresponding to a fixed $(s, a)$ from the online environment.

To address this issue, we follow prior works [18, 23] and incorporate the conditioning variables directly into the transport cost. Specifically, consider pairs of transition samples $(s_{\mathrm{off}}, a_{\mathrm{off}}, s'_{\mathrm{off}})$ and $(s_{\mathrm{on}}, a_{\mathrm{on}}, s'_{\mathrm{on}})$. We define the transport cost as

$$c\big((s_{\mathrm{off}}, a_{\mathrm{off}}, s'_{\mathrm{off}}), (s_{\mathrm{on}}, a_{\mathrm{on}}, s'_{\mathrm{on}})\big) = \|s'_{\mathrm{off}} - s'_{\mathrm{on}}\|_2^2 + \eta\big(\|s_{\mathrm{off}} - s_{\mathrm{on}}\|_2^2 + \|a_{\mathrm{off}} - a_{\mathrm{on}}\|_2^2\big),$$

where $\eta > 0$ controls the strength of alignment between conditioning variables.

Let $q^*$ denote the optimal coupling between the empirical distributions of $(s_{\mathrm{off}}, a_{\mathrm{off}}, s'_{\mathrm{off}})$ and $(s_{\mathrm{on}}, a_{\mathrm{on}}, s'_{\mathrm{on}})$ under the above cost. The online flow is trained using the objective

$$\mathcal{L}_{\mathrm{on}}(\phi) = \mathbb{E}_{((s_{\mathrm{off}}, a_{\mathrm{off}}, s'_{\mathrm{off}} \doteq x_1), (s_{\mathrm{on}}, a_{\mathrm{on}}, s'_{\mathrm{on}} \doteq x_2)) \sim q^*, t \sim \mathcal{U}[1,2]} \left[ \|v_\phi(x_t, t, s_{\mathrm{on}}, a_{\mathrm{on}}) - (s'_{\mathrm{on}} - s'_{\mathrm{off}})\|^2 \right], \tag{3}$$

where $x_t$ is the linear interpolation between $x_1$ and $x_2$.

**Proposition 3.3** (Informal; shared-conditioning coupling is $W_2$-optimal). *Assume that, when training the online flow in Eq. (3), the marginals over the conditioning variables are matched, i.e., $p_{\mathrm{off}}(s, a) = p_{\mathrm{on}}(s, a)$. Then, for any fixed $(s, a) \in \mathcal{S} \times \mathcal{A}$, as $\eta \to \infty$ and the minibatch size used to compute the OT coupling tends to infinity, the expected squared displacement induced by the composite flow recovers the squared 2-Wasserstein distance between the offline and online transition distributions:*

$$\mathbb{E}_{x_0 \sim \mathcal{N}(0, \mathbf{I})} \left[ \|\psi_\theta^{\mathrm{off}}(x_0, 1|s, a) - \psi_\phi^{\mathrm{on}}(\psi_\theta^{\mathrm{off}}(x_0, 1|s, a), 2|s, a)\|_2^2 \right] \longrightarrow W_2^2(p_{\mathrm{off}}(\cdot|s, a), p_{\mathrm{on}}(\cdot|s, a)).$$

**Monte Carlo Estimator.** This result yields a practical Monte Carlo estimator of the dynamics gap $\Delta(s, a)$:

$$\widehat{\Delta}(s, a) = \left( \frac{1}{M} \sum_{j=1}^{M} \|\psi_\theta^{\mathrm{off}}(x_0^{(j)}, 1|s, a) - \psi_\phi^{\mathrm{on}}(\psi_\theta^{\mathrm{off}}(x_0^{(j)}, 1|s, a), 2|s, a)\|_2^2 \right)^{1/2}, \quad x_0^{(j)} \sim \mathcal{N}(0, \mathbf{I}). \tag{4}$$

*Remark* 3.4 (Enforcing the shared marginal condition). The marginal condition $p_{\mathrm{off}}(s, a) = p_{\mathrm{on}}(s, a)$ can be enforced by construction. In practice, we construct the empirical distribution of offline triples $(s_{\mathrm{off}}, a_{\mathrm{off}}, s'_{\mathrm{off}})$ by first sampling $(s_{\mathrm{off}}, a_{\mathrm{off}})$ from the **online** dataset $\mathcal{D}_{\mathrm{on}}$, and then generating $s'_{\mathrm{off}} \sim p_{\mathrm{off}}(\cdot|s_{\mathrm{off}}, a_{\mathrm{off}})$ using the pretrained offline flow. Separately, we sample online transitions $(s_{\mathrm{on}}, a_{\mathrm{on}}, s'_{\mathrm{on}})$ from $\mathcal{D}_{\mathrm{on}}$. As a result, the two empirical measures used in the OT coupling have matched $(s, a)$-marginals in expectation, while their conditional transitions differ. The complete training algorithms of the offline flow and the online flow are in Appendix C.

## 3.2  Data Collection at High Dynamics Gap Region

As discussed in Section 2.1, using behavior cloning and augmenting the replay buffer with low dynamics gap offline data can alleviate performance drop from distribution shift. However, relying solely on such data may limit state-action coverage and hinder policy learning. To address this, we propose a new data collection strategy.

At each training iteration, we construct the replay buffer by selectively incorporating offline transitions with small estimated dynamics gap:

$$\mathcal{B} = \left\{ (s,a) \in \mathcal{D}_{\text{off}} : \widehat{\Delta}(s,a) \leq \tau \right\} \cup \mathcal{D}_{\text{on}}, \tag{5}$$

where $\tau$ is a predefined threshold. To improve data diversity and encourage better generalization, we actively explore regions with high dynamics gap—areas likely underrepresented in the buffer due to the dominance of low-gap samples. We adopt an optimistic exploration policy that selects actions by

$$a = \arg\max_{a \in \mathcal{A}} \left[ Q(s,a) + \beta \, \widehat{\Delta}(s,a) \right], \tag{6}$$

where $\beta$ is a hyperparameter that trades off return and exploration of underexplored dynamics. To ensure sufficient coverage during training, we can further incorporate stochasticity by adding small perturbations to the selected actions [17] or following a stochastic policy [19] consistent with widely used deep RL frameworks.

**Theorem 3.5** (Large Dynamics Gap Exploration Reduces Performance Gap). *Compared to behavior cloning policy $\pi_{\text{bc}}$ on the offline dataset, training a policy $\hat{\pi}$ by replacing all offline samples with a dynamics gap exceeding $\kappa$ (as estimated by the composite flow) with online environment samples can reduce the performance gap to the optimal online policy $\pi_{on}^*$ with high probability by*

$$\frac{2L_r(1+\gamma)}{(1-\gamma)(1-\gamma L_p)} \Big( \Delta_{W_2} - \kappa - \sqrt{\left( C_0 + C_1 \, W_2(\hat{p}_{\text{off}}, p_{\text{on}}) \right) \Gamma_{N_{\text{on}}, \delta}} \Big). \tag{7}$$

*Here $L_r$ and $L_P$ are the Lipschitz constants for the reward and transition function, respectively. $\gamma L_p < 1$. $\Delta_{W_2} := \sup_{s,a} W_2\big(p_{\text{off}}(\cdot|s,a), \, p_{\text{on}}(\cdot|s,a)\big)$ is the largest dynamics gap. $C_0$ and $C_1$ are two constants. $\Gamma_{N_{\text{on}}, \delta} := \mathfrak{R}_{N_{\text{on}}}(\mathcal{H}) + \sqrt{\frac{\log(1/\delta)}{N_{\text{on}}}}$, where $\mathcal{H}$ is the same as in Theorem 3.1 and $N_{\text{on}}$ is the number of samples used to train the online flow.*

From Theorem 3.5, exploration in regions with high dynamics gap can reduce the performance gap relative to the optimal policy. The parameter $\kappa$ serves as a threshold: the bound holds when the policy is trained without offline samples whose dynamics gap exceeds $\kappa$. As $\beta$ increases, more samples are collected from high-gap regions, which decreases $\kappa$. This increases performance gap reduction, resulting in a policy with higher expected return.

### 3.3  Practical Implementation

Our method can be instantiated using standard actor-critic algorithms with a critic $Q_\varsigma(s,a)$ and a policy $\pi_\varphi(a|s)$. To incorporate the dynamics gap, we apply rejection sampling to retain a fixed percentage of offline transitions with the lowest estimated gap in each iteration. The critic is trained by minimizing

$$\mathcal{L}_Q = \mathbb{E}_{(s,a,s') \sim D_{\text{on}}}[(Q_\varsigma(s,a) - y)^2] + \mathbb{E}_{(s,a,s') \sim D_{\text{off}}}[\mathbf{1}(\widehat{\Delta}(s,a) \leq \widehat{\Delta}_{\xi\%})(Q_\varsigma(s,a) - y)^2], \tag{8}$$

where $\widehat{\Delta}_{\xi\%}$ is the $\xi$-quantile of the estimated dynamics gap in the offline minibatch. The target value is $y = r + \gamma Q_\varsigma(s',a') + \beta \widehat{\Delta}(s,a)$, with $a' \sim \pi_\varphi(\cdot|s')$.

The policy is updated by combining policy improvement with a behavior cloning regularizer, using the objective

$$\mathcal{L}_\pi = \mathbb{E}_{s \sim \mathcal{D}_{\text{off}} \cup \mathcal{D}_{\text{on}}, \, a \sim \pi_\varphi(\cdot|s)} \big[ Q_\varsigma(s,a) \big] - \omega \, \mathbb{E}_{(s,a) \sim \mathcal{D}_{\text{off}}, \, \tilde{a} \sim \pi_\varphi(\cdot|s)} \big[ \|a - \tilde{a}\|_2^2 \big], \tag{9}$$

where $\omega > 0$ controls the trade-off between maximizing the estimated $Q$-value and staying close to the offline behavior. The behavior cloning term encourages the policy's sampled action $\tilde{a}$ to match the offline action $a$, as motivated by Lemma 2.2.

The pseudocode of COMPFLOW, instantiated with Soft Actor-Critic (SAC) [19], is presented in Appendix D.

# 4 Related Work

**Online RL with offline dataset.** Online RL often requires extensive environment interactions [54, 68], which can be costly or impractical in real-world settings. To improve sample efficiency, Offline-to-Online RL leverages pre-collected offline data to bootstrap online learning [43]. A typical two-phase approach trains an initial policy offline, then fine-tunes it online [32, 43, 44]. However, conservative strategies used to mitigate distributional shift, such as pessimistic value estimation [31], can result in suboptimal initial policies and limit effective exploration [40, 44]. To resolve this tension, recent methods propose ensemble-based pessimism [32], value calibration [44], optimistic action selection [32], and policy expansion [70]. Others directly incorporate offline data into the replay buffer of off-policy algorithms [3], improving stability via ensemble distillation and layer normalization. However, these approaches typically assume that the offline dataset is generated under the same transition dynamics as the online environment. In contrast, our work explicitly accounts for dynamics shift between the offline data and the online environment.

**RL with dynamics shift.** Our work is related to cross-domain RL, where the source and target domains share the same observation and action spaces but differ in transition dynamics. Prior work has addressed such discrepancies via system identification [5, 8, 10], domain randomization [41, 56, 61], imitation learning [20, 24], and meta-RL [42, 50], often assuming shared environment distributions [58] or requiring expert demonstrations. More recent work has relaxed these assumptions. One line of research focuses on *reward modification*, which adjusts the reward function to penalize source transitions that are unlikely under the target dynamics [12, 35], or down-weights value estimates in regions with high dynamics gap [46]. Another line of work explores *data filtering*, which selects only source transitions with low estimated dynamics gap [65], using metrics such as transition probability ratios [12], mutual information [64], value inconsistency [65], or representation-based KL divergence [37]. However, these metrics can become unstable or ill-defined when the transition dynamics have different supports, and value-based methods are prone to instability caused by bootstrapping bias.

In contrast, our approach leverages the theoretical connection between optimal transport and flow matching to estimate the dynamics gap in a principled manner. A closely related work by [39] also applies optimal transport to quantify the dynamics gap. However, their setting assumes both the source and target domains are offline, and their method estimates the gap based on the concatenated tuple $(s, a, s')$ observed in offline data, rather than comparing conditional transition distributions. As a result, their metric does not accurately capture the gap in transition dynamics and cannot be used to compute exploration bonuses, which require gap estimation conditioned on a given state-action pair.

**RL with diffusion and flow models.** Diffusion models [21, 28, 57] and flow matching [11, 34] have emerged as powerful generative tools capable of modeling complex, high-dimensional distributions. Their application in RL is consequently expanding. Researchers have employed these generative models for various tasks, including planning and trajectory synthesis [22], representing expressive multimodal policies [7, 51], providing behavior regularization [6], or augmenting training datasets with synthesized experiences. While these works often focus on policy learning or modeling dynamics within a single environment, our approach targets the transfer learning setting. Specifically, we address scenarios characterized by a *dynamics gap* between the offline data and the online environment. We utilize Flow Matching, leveraging its connection to optimal transport, to estimate this gap.

# 5 Experiments

In this section[2], we first evaluate our approach across a range of environments in Gym-MuJoCo that exhibit different types of dynamics shifts. We then conduct ablation and hyperparameter studies to better understand the design choices and behavior of COMPFLOW. Finally, we assess the effectiveness of our method in a real-world inspired wildlife conservation task.

## 5.1 Gym-MuJoCo

### 5.1.1 Experimental Setup

**Tasks and datasets.** We evaluate our algorithm under three types of dynamics shifts, namely morphology, kinematic and friction changes, across three OpenAI Gym locomotion tasks: HalfCheetah,

---

[2]Our code is available at `https://github.com/Haichuan23/CompositeFlow`

| Dataset | Task Name | SAC | BC-SAC | H2O | BC-VGDF | BC-PAR | Ours |
|---|---|---|---|---|---|---|---|
| MR | HalfCheetah (Morphology) | $1457 \pm 89$ | $2495 \pm 43$ | $1430 \pm 408$ | $2765 \pm 124$ | $1790 \pm 91$ | $3119 \pm 107$ |
| MR | HalfCheetah (Kinematic) | $2255 \pm 197$ | $4868 \pm 186$ | $4257 \pm 609$ | $4392 \pm 403$ | $4179 \pm 441$ | $5189 \pm 262$ |
| MR | HalfCheetah (Friction) | $2069 \pm 184$ | $7799 \pm 157$ | $6397 \pm 673$ | $7829 \pm 821$ | $8056 \pm 512$ | $8241 \pm 180$ |
| MR | Hopper (Morphology) | $364 \pm 82$ | $346 \pm 4$ | $361 \pm 18$ | $348 \pm 21$ | $354 \pm 25$ | $355 \pm 6$ |
| MR | Hopper (Kinematic) | $737 \pm 547$ | $1024 \pm 0$ | $1025 \pm 0$ | $1024 \pm 0$ | $1024 \pm 1$ | $1024 \pm 1$ |
| MR | Hopper (Friction) | $234 \pm 4$ | $228 \pm 2$ | $229 \pm 1$ | $230 \pm 3$ | $232 \pm 5$ | $280 \pm 27$ |
| MR | Walker2D (Morphology) | $253 \pm 60$ | $598 \pm 475$ | $1014 \pm 193$ | $672 \pm 576$ | $458 \pm 151$ | $1094 \pm 791$ |
| MR | Walker2D (Kinematic) | $152 \pm 22$ | $2973 \pm 185$ | $1967 \pm 851$ | $1586 \pm 923$ | $948 \pm 131$ | $1568 \pm 1315$ |
| MR | Walker2D (Friction) | $301 \pm 9$ | $311 \pm 5$ | $296 \pm 14$ | $302 \pm 12$ | $321 \pm 26$ | $344 \pm 20$ |
| M | HalfCheetah (Morphology) | $1467 \pm 89$ | $1522 \pm 72$ | $1720 \pm 273$ | $1829 \pm 345$ | $1427 \pm 196$ | $2282 \pm 287$ |
| M | HalfCheetah (Kinematic) | $2316 \pm 92$ | $5451 \pm 195$ | $5019 \pm 773$ | $4972 \pm 381$ | $5243 \pm 120$ | $5593 \pm 44$ |
| M | HalfCheetah (Friction) | $2028 \pm 238$ | $7108 \pm 1001$ | $6968 \pm 846$ | $6802 \pm 956$ | $7800 \pm 525$ | $7871 \pm 238$ |
| M | Hopper (Morphology) | $396 \pm 60$ | $436 \pm 45$ | $410 \pm 8$ | $406 \pm 52$ | $418 \pm 13$ | $604 \pm 173$ |
| M | Hopper (Kinematic) | $724 \pm 535$ | $1022 \pm 1$ | $970 \pm 98$ | $934 \pm 43$ | $1020 \pm 3$ | $1023 \pm 2$ |
| M | Hopper (Friction) | $229 \pm 5$ | $232 \pm 1$ | $228 \pm 4$ | $229 \pm 4$ | $233 \pm 5$ | $300 \pm 66$ |
| M | Walker2D (Morphology) | $301 \pm 177$ | $457 \pm 317$ | $577 \pm 201$ | $584 \pm 219$ | $431 \pm 177$ | $886 \pm 372$ |
| M | Walker2D (Kinematic) | $258 \pm 174$ | $1966 \pm 1155$ | $1965 \pm 568$ | $1921 \pm 928$ | $806 \pm 278$ | $2039 \pm 936$ |
| M | Walker2D (Friction) | $301 \pm 9$ | $286 \pm 54$ | $298 \pm 45$ | $289 \pm 47$ | $308 \pm 24$ | $320 \pm 31$ |
| ME | HalfCheetah (Morphology) | $1392 \pm 238$ | $1195 \pm 241$ | $1147 \pm 169$ | $1072 \pm 102$ | $1207 \pm 53$ | $1485 \pm 67$ |
| ME | HalfCheetah (Kinematic) | $2323 \pm 97$ | $4211 \pm 262$ | $5143 \pm 330$ | $4603 \pm 498$ | $4399 \pm 164$ | $5750 \pm 84$ |
| ME | HalfCheetah (Friction) | $1950 \pm 312$ | $4185 \pm 732$ | $2140 \pm 733$ | $4078 \pm 1032$ | $4989 \pm 500$ | $5596 \pm 1557$ |
| ME | Hopper (Morphology) | $359 \pm 75$ | $349 \pm 47$ | $444 \pm 15$ | $357 \pm 63$ | $407 \pm 28$ | $462 \pm 89$ |
| ME | Hopper (Kinematic) | $724 \pm 535$ | $1024 \pm 1$ | $1031 \pm 3$ | $1022 \pm 3$ | $1027 \pm 8$ | $1022 \pm 2$ |
| ME | Hopper (Friction) | $229 \pm 4$ | $230 \pm 3$ | $232 \pm 5$ | $230 \pm 6$ | $232 \pm 8$ | $266 \pm 70$ |
| ME | Walker2D (Morphology) | $228 \pm 51$ | $429 \pm 117$ | $1103 \pm 444$ | $502 \pm 301$ | $380 \pm 231$ | $648 \pm 180$ |
| ME | Walker2D (Kinematic) | $386 \pm 184$ | $850 \pm 953$ | $1514 \pm 782$ | $1204 \pm 734$ | $755 \pm 268$ | $1511 \pm 1206$ |
| ME | Walker2D (Friction) | $266 \pm 66$ | $240 \pm 114$ | $258 \pm 8$ | $245 \pm 51$ | $242 \pm 24$ | $326 \pm 26$ |
| | **Average Return** | **878** | **1920** | **1783** | **1868** | **1803** | **2193** |

Table 1: Comparison of return under different dynamics shift scenarios and dataset types after 40K environment interactions. MR = Medium Replay, M = Medium, ME = Medium Expert. A cell is green if the method has the highest mean and improves over the second best by at least 2%. Cells within 2% of the top mean are marked in yellow.

Hopper, and Walker2d [4]. Each experiment involves an offline environment and an online environment with modified transition dynamics following [38]. Morphology shifts alter the sizes of body parts, kinematic shifts impose constraints on joint angles, and friction shifts modify the static, dynamic, and rolling friction coefficients. For each task, we use three D4RL source datasets: medium, medium replay, and medium expert, which capture varying levels of data quality [16]. Additional environment details are in Appendix K.

**Baselines.** We compare COMPFLOW against the following baselines: BC-SAC extends SAC by incorporating both offline and online data, with a behavior cloning (BC) term for the offline data. H2O [46] penalizes Q-values for state-action pairs with large dynamics gaps. BC-VGDF [65] selects offline transitions with value targets consistent with the online environment and adds a BC term. BC-PAR [37] applies a reward penalty based on representation mismatch between offline and online transitions, also including a BC term. Implementation details are provided in Appendix K.

### 5.1.2 Main Results

We evaluate each algorithm by measuring its return in the target environment after 40K environment interactions and 400K gradient updates, reflecting a limited-interaction setting. The results are reported in Table 1. Our key findings are summarized as follows:

**(1)** COMPFLOW consistently outperforms recent off-dynamics RL baselines across diverse offline dataset qualities and dynamics shifts. It achieves the best return on **19/27** tasks and ties for best on **5** more. On average, COMPFLOW reaches a score of **2193**, compared to **1920** for the strongest baseline (BC-SAC), a **14.2%** relative improvement. Additional results under more severe dynamics shifts in Appendix J show the same trend.

**(2)** Leveraging offline data substantially improves performance: on average, COMPFLOW improves over SAC by **149.8%**. Moreover, COMPFLOW outperforms BC-SAC, which directly uses all offline data, on **23/27** tasks and matches it on the remaining **3**, highlighting the benefit of our dynamics-gap-aware design.

**(3)** We also find that many recent baselines perform similarly to BC-SAC, consistent with [38], suggesting limited ability to effectively exploit offline data under dynamics shift. One reason is that methods such as H2O and BC-PAR estimate the dynamics gap using KL divergence or mutual information, which can be ill-defined under large shifts or mismatched support. BC-VGDF instead filters data based on value estimates, which is often harder due to bootstrapping bias and target non-stationarity. In contrast, COMPFLOW models transition dynamics with flow matching and estimates the dynamics gap via the Wasserstein distance through its connection to optimal transport, yielding a more principled and robust criterion.

### 5.1.3 Ablation and Hyperparameter Analysis

**Composite Flow.** We first evaluate the effectiveness of the proposed composite flow matching design. For comparison, we train a direct flow model on the target environment initialized from a Gaussian prior. We compute the mean squared error (MSE) on a held-out 10% validation set and report the average MSE across different epochs during RL training. As shown in Figure 3, the composite flow significantly reduces the MSE of the transition dynamics in the environment. This improvement stems from its

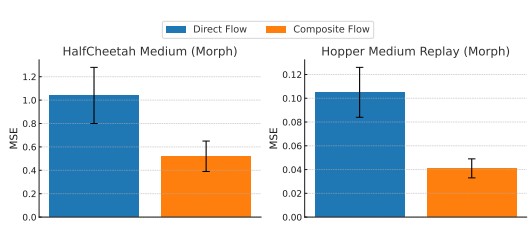

Figure 3: Comparison of MSE between direct flow and composite flow.

ability to reuse structural knowledge learned from the abundant offline data. These empirical findings support the theoretical insight presented in Theorem 3.1 that our composite flow can improve the generalization ability.

**Impact of data selection ratio** $\xi\%$. The data selection ratio $\xi$ decides how many offline data in a sampled batch can be shared for policy training. A larger $\xi$ indicates that more offline data will be admitted. To examine its influence, We sweep $\xi$ across $\{70, 50, 30, 20\}$. The results is shown in Figure 4. Although the optimal $\xi$ seems task dependent, moderate values of $\xi$ (e.g., 30 or 50) generally yield good performance across tasks, striking a balance between leveraging useful offline data and avoiding high dynamics mismatch. When $\xi$ is too large (e.g., 70), performance often degrades, particularly in Walker Medium (Morph) and Walker Medium Replay (Morph), likely due to the inclusion of low-quality transitions with large dynamics gap. HalfCheetah consistently achieves higher returns compared to other domains, suggesting that knowledge transfer and policy learning in this environment are easier. Consequently, overly conservative filtering (e.g., $\xi = 20$) may exclude valuable offline data, leading to slower learning. In this case, allowing more offline data (e.g., $\xi = 70$) appears beneficial.

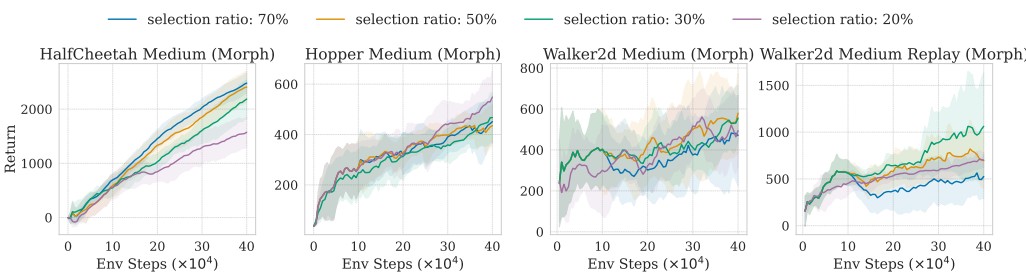

Figure 4: Comparison of return under different data selection ratios across tasks.

**Impact of exploration strength** $\beta$. $\beta$ controls the strength of exploration toward regions with large dynamics gap. A large $\beta$ indicates higher incentive to explore such regions. As shown in Figure 5, the effect of $\beta$ on return is task-dependent, possibly due to differences in the underlying MDPs. In the Friction tasks, we observe that larger exploration bonuses lead to higher asymptotic returns. This suggests that incentivizing exploration helps the algorithm discover high-reward regions more effectively. In contrast, in the Morphology tasks, moderate values of $\beta$ typically outperform both smaller and larger values. This indicates that excessive exploration may not be effective in some tasks. While the optimal $\beta$ varies by task, one trend is consistent: across all five experiments, the setting with no exploration ($\beta = 0$) consistently ranks as the lowest or the second lowest performers. This

empirical finding confirms the importance of the exploration bonus and provides evidence supporting our algorithmic design.

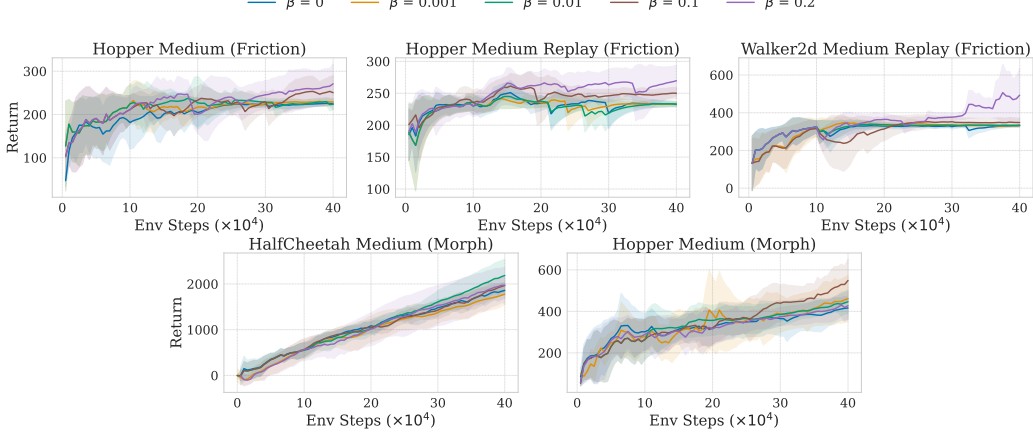

Figure 5: Comparison of return under different exploration strengths across tasks.

## 5.2 Patrol Policy Learning for Wildlife Conservation

We evaluate COMPFLOW in a wildlife conservation setting, where the goal is to learn a patrol policy that allocates limited ranger effort to reduce poaching. We adopt the simulation environment of Xu et al. [67], which represents the park as a $1 \times 1$ km spatial grid. At each time step, the agent allocates a distribution of patrol effort across cells under a fixed budget. The state, including wildlife density and poaching risk, evolves according to dynamics driven by both poacher behavior and patrol deployment, and the per-step reward reflects the expected number of animals preserved through deterrence and spatial coverage.

We assume access to an offline dataset collected under a previous policy in Murchison Falls National Park, and aim to leverage it to learn an improved patrol policy for Queen Elizabeth National Park with only limited online interactions. Since ecological conditions and poacher behavior differ across parks, the transition dynamics are mismatched. As shown in Figure 6, COMPFLOW achieves the highest reward, outperforming the best baseline, BC-PAR, by **8.8%**. It also improves over training from scratch with SAC by **20.8%** under the same interaction budget. These gains are particularly valuable in large protected areas where ranger capacity is constrained: Murchison Falls spans roughly 3,900 km$^2$, while Queen Elizabeth covers about 1,980 km$^2$, making data-efficient learning essential for effective protection.

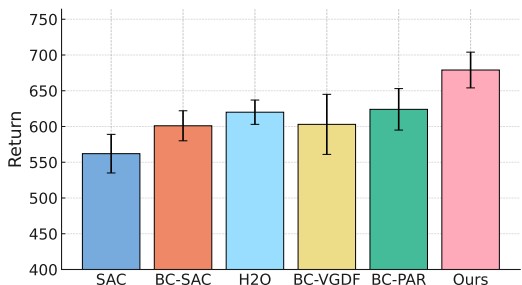

Figure 6: Reward on the wildlife conservation task.

## 6 Conclusion and Limitations

In this paper, we proposed COMPFLOW, a new method for estimating the dynamics gap in reinforcement learning with shifted-dynamics data. COMPFLOW leverages the theoretical connection between flow matching and optimal transport. To address data scarcity in the online environment, we adopt a composite flow structure that builds the online flow model on top of the output distribution from the offline flow. This composite formulation improves generalization and enables the use of Wasserstein distance between offline and online transitions as a robust measure of the dynamics gap. Using this pricnipled estimation, we further encourage the policy to explore regions with high dynamics gap and provide a theoretical analysis of the benefits. Empirically, we demonstrate that COMPFLOW consistently outperforms or matches state-of-the-art baselines across diverse RL tasks with varying types of dynamics shift.

Due to the space limit, we discuss the limitations of COMPFLOW in Appendix A.

## Acknowledgement

We are thankful to the Uganda Wildlife Authority for granting us access to incident data from Murchison Falls and Queen Elizabeth National Park. We also thank the anonymous reviewers for their valuable feedback. This work was supported by ONR MURI N00014-24-1-2742.

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

# Appendix for Composite Flow Matching for Reinforcement Learning with Shifted-Dynamics Data

## A    Limitations

(1) COMPFLOW is currently limited to settings where the offline data and the online environment share the same state and action spaces. Future work could explore estimating the dynamics gap in a shared latent embedding space to relax this assumption [71]. (2) Our evaluation is conducted entirely in simulated environments; applying COMPFLOW to real-world scenarios remains an important direction for future work. (3) We have not yet incorporated the uncertainty in estimating the dynamics gap. Future work may explore how to quantify this uncertainty and use it to improve both data filtering and exploration strategies [27, 33].

## B    Broader Impacts

Our work aims to make reinforcement learning more practical in real-world domains such as healthcare, robotics, and conservation, where online interaction is often costly, limited, or unsafe. By addressing the dynamics shift between offline data and the online environment, our method enables more reliable and sample-efficient policy learning. This capability supports safer deployment in high-stakes applications, including clinical decision support and adaptive anti-poaching strategies. Nonetheless, we emphasize that policies trained on historical data should be applied with caution,

as misaligned dynamics or biased datasets may lead to unintended consequences if not carefully validated.

## C  Algorithms of Training Optimal Transport Flow Matching

The full training algorithm of Optimal Transport Flow Matching is given in Algorithm 1.

---

**Algorithm 1** Training OT Flow Matching (OT-FM)

---

**Require:** Dataset $\mathcal{D}$; batch size $k$; learning rate $\mathsf{lr}$; vector field $v_\theta$
**Ensure:** Trained parameters $\theta$
1: **while** not converged **do**
2:     **(Sample)** Draw latent codes $(x_0^{(i)})_{i=1}^k \sim \mathcal{N}(0, \mathbf{I})$
3:     Draw data points $(x_1^{(j)})_{j=1}^k \sim \mathcal{D}$
4:     **(OT coupling)** Set $C_{ij} \leftarrow \|x_0^{(i)} - x_1^{(j)}\|_2^2$ for all $(i,j) \in [k] \times [k]$
5:     Compute an optimal transport plan

$$A \leftarrow \arg\min_{A \in \mathcal{B}_k} \langle A, C \rangle, \qquad \mathcal{B}_k := \{A \in \mathbb{R}_+^{k \times k} : A\mathbf{1} = \tfrac{1}{k}\mathbf{1}, \ A^\top \mathbf{1} = \tfrac{1}{k}\mathbf{1}\}.$$

                                                    ▷ $A$ is a (scaled) doubly-stochastic coupling
6:     Sample index pairs $(i_\ell, j_\ell)_{\ell=1}^k$ i.i.d. from the categorical distribution on $[k] \times [k]$ with probabilities $A_{ij}$
7:     Sample times $(t_\ell)_{\ell=1}^k \sim \mathcal{U}[0,1]$
8:     **for** $\ell = 1$ to $k$ **do**
9:         $x_{t_\ell}^{(\ell)} \leftarrow (1 - t_\ell)\, x_0^{(i_\ell)} + t_\ell\, x_1^{(j_\ell)}$
10:        $\Delta^{(\ell)} \leftarrow x_1^{(j_\ell)} - x_0^{(i_\ell)}$
11:        $\ell_\ell \leftarrow \big\|v_\theta(x_{t_\ell}^{(\ell)}, t_\ell) - \Delta^{(\ell)}\big\|_2^2$
12:    **end for**
13:    $\mathcal{L} \leftarrow \frac{1}{k}\sum_{\ell=1}^k \ell_\ell$
14:    $\theta \leftarrow \theta - \mathsf{lr}\, \nabla_\theta \mathcal{L}$
15: **end while**
16: **return** $\theta$

---

## D  Algorithms of Training Offline and Online Flows

The full training algorithms of the offline flow and online flow are given in Algorithm 2 and Algorithm 3 respectively.

---

**Algorithm 2** Training the Offline Flow via Flow Matching

---

**Require:** Offline dataset $\mathcal{D}_{\mathrm{off}}$; batch size $k$; offline vector field $v_\theta$; learning rate $\mathsf{lr}$
**Ensure:** Trained offline flow parameters $\theta$
1: **while** not converged **do**
2:     Sample transitions $(s^{(i)}, a^{(i)}, s'^{(i)})_{i=1}^k \sim \mathcal{D}_{\mathrm{off}}$
3:     Set $x_1^{(i)} \leftarrow s'^{(i)}$ for all $i \in [k]$
4:     Sample latent codes $(x_0^{(i)})_{i=1}^k \sim \mathcal{N}(0, \mathbf{I})$
5:     Sample times $(t_i)_{i=1}^k \sim \mathcal{U}[0,1]$
6:     **for** $i = 1$ to $k$ **do**
7:         $x_{t_i}^{(i)} \leftarrow (1 - t_i)\, x_0^{(i)} + t_i\, x_1^{(i)}$
8:         $\Delta^{(i)} \leftarrow x_1^{(i)} - x_0^{(i)}$
9:         $\ell_i \leftarrow \big\|v_\theta(x_{t_i}^{(i)}, t_i, s^{(i)}, a^{(i)}) - \Delta^{(i)}\big\|_2^2$
10:    **end for**
11:    $\mathcal{L} \leftarrow \frac{1}{k}\sum_{i=1}^k \ell_i$
12:    $\theta \leftarrow \theta - \mathsf{lr}\, \nabla_\theta \mathcal{L}$
13: **end while**
14: **return** $\theta$

---

**Algorithm 3** Training the Online Flow via Optimal Transport

---

**Require:** Pre-trained offline flow sampler $\psi_\theta(x, t \mid s, a)$; online replay buffer $\mathcal{D}_{\mathrm{on}}$; batch size $k$; regularization weight $\eta > 0$; learning rate $\mathsf{lr}$; initialized online vector field $v_\phi$

**Ensure:** Trained online flow parameters $\phi$

1: **while** not converged **do**
2:     **(Offline-synthetic batch)** Sample $(s_{\mathrm{off}}^{(i)}, a_{\mathrm{off}}^{(i)})_{i=1}^k \sim \mathcal{D}_{\mathrm{on}}$
3:     Sample latent codes $(x_0^{(i)})_{i=1}^k \sim \mathcal{N}(0, \mathbf{I})$
4:     **for** $i = 1$ to $k$ **do**
5:         $x_1^{(i)} \leftarrow s_{\mathrm{off}}'^{(i)} \leftarrow \psi_\theta(x_0^{(i)}, 1 | s_{\mathrm{off}}^{(i)}, a_{\mathrm{off}}^{(i)})$
6:     **end for**
7:     **(Online batch)** Sample $(s_{\mathrm{on}}^{(j)}, a_{\mathrm{on}}^{(j)}, s_{\mathrm{on}}'^{(j)})_{j=1}^k \sim \mathcal{D}_{\mathrm{on}}$
8:     $x_2^{(j)} \leftarrow s_{\mathrm{on}}'^{(j)}$ for all $j \in [k]$
9:     **(OT coupling)** For all $(i, j) \in [k] \times [k]$, set

$$C_{ij} \leftarrow \|x_1^{(i)} - x_2^{(j)}\|_2^2 + \eta\big(\|s_{\mathrm{off}}^{(i)} - s_{\mathrm{on}}^{(j)}\|_2^2 + \|a_{\mathrm{off}}^{(i)} - a_{\mathrm{on}}^{(j)}\|_2^2\big).$$

10:    Compute an optimal transport plan

$$A \leftarrow \arg \min_{A \in \mathcal{B}_k} \langle A, C \rangle, \qquad \mathcal{B}_k := \{A \in \mathbb{R}_+^{k \times k} : A\mathbf{1} = \tfrac{1}{k}\mathbf{1}, \ A^\top \mathbf{1} = \tfrac{1}{k}\mathbf{1}\}.$$

                                          $\triangleright$ $A$ is a (scaled) doubly-stochastic coupling
11:    Sample index pairs $(i_\ell, j_\ell)_{\ell=1}^k$ i.i.d. from the categorical distribution on $[k] \times [k]$ with probabilities $A_{ij}$
12:    Sample times $(t_\ell)_{\ell=1}^k \sim \mathcal{U}[1, 2]$
13:    **for** $\ell = 1$ to $k$ **do**
14:       $x_{t_\ell}^{(\ell)} \leftarrow (2 - t_\ell)\, x_1^{(i_\ell)} + (t_\ell - 1)\, x_2^{(j_\ell)}$
15:       $\Delta^{(\ell)} \leftarrow x_2^{(j_\ell)} - x_1^{(i_\ell)}$
16:       $\ell_\ell \leftarrow \big\|v_\phi(x_{t_\ell}^{(\ell)}, t_\ell, s_{\mathrm{on}}^{(j_\ell)}, a_{\mathrm{on}}^{(j_\ell)}) - \Delta^{(\ell)}\big\|_2^2$
17:    **end for**
18:    $\mathcal{L} \leftarrow \frac{1}{k} \sum_{\ell=1}^k \ell_\ell$
19:    $\phi \leftarrow \phi - \mathsf{lr}\, \nabla_\phi \mathcal{L}$
20: **end while**
21: **return** $\phi$

---

## E   Algorithm of COMPFLOW built on Soft-Actor-Critic

The full training algorithm of COMPFLOW built on SAC is given in Algorithm 4.

**Algorithm 4** COMPFLOW built on Soft Actor-Critic (SAC)

---

1: **Input:** Offline dataset $\mathcal{D}_{\text{off}}$, online environment $\mathcal{M}_{\text{on}}$, max interaction steps $T_{\max}$, update frequency train_freq, offline selection ratio $\xi \in (0, 1]$, gap reward scale $\beta$, batch size $B$, behavior cloning weight $\omega$, warmup steps warmup_steps, gradient steps per interaction $K$, learning rate lr, target update rate $\varpi$, discount $\gamma$, entropy temperature $\alpha$.

2: **Initialize:** Policy $\pi_\varphi$, critics $\{Q_{\varsigma_i}\}_{i=1,2}$, target critics $\{Q_{\varsigma_i}^{\text{tgt}}\}_{i=1,2} \leftarrow \{Q_{\varsigma_i}\}_{i=1,2}$, online replay buffer $\mathcal{D}_{\text{on}} \leftarrow \emptyset$.

3: Pretrain offline flow $\psi_\theta(x, t \mid s, a)$ on $\mathcal{D}_{\text{off}}$ via Algorithm 2.

4: **for** $t = 1$ to $T_{\max}$ **do**

5:     Interact with $\mathcal{M}_{\text{on}}$ using $\pi_\varphi$ and store transition $(s, a, r, s')$ into $\mathcal{D}_{\text{on}}$.         $\triangleright a \sim \pi_\varphi(\cdot \mid s)$, $(r, s') \sim \mathcal{M}_{\text{on}}(s, a)$

6:     **if** $t >$ warmup_steps **and** $t \bmod$ train_freq $= 0$ **then**

7:         Train online flow $\psi_\phi(x, t \mid s, a)$ on $\mathcal{D}_{\text{on}}$ via Algorithm 3.

8:     **end if**

9:     **for** $k = 1$ to $K$ **do**

10:         Sample offline minibatch $b_{\text{off}} = \{(s_{\text{off}}^{(i)}, a_{\text{off}}^{(i)}, r_{\text{off}}^{(i)}, s_{\text{off}}'^{(i)})\}_{i=1}^{B} \sim \mathcal{D}_{\text{off}}$.

11:         Sample online minibatch $b_{\text{on}} = \{(s_{\text{on}}^{(i)}, a_{\text{on}}^{(i)}, r_{\text{on}}^{(i)}, s_{\text{on}}'^{(i)})\}_{i=1}^{B} \sim \mathcal{D}_{\text{on}}$.

12:         Estimate dynamics gap $\widehat{\Delta}(s_{\text{off}}^{(i)}, a_{\text{off}}^{(i)})$ for each $(s_{\text{off}}^{(i)}, a_{\text{off}}^{(i)})$ via Eq. (4).

13:         Select the lowest-gap subset $\tilde{b}_{\text{off}} \subset b_{\text{off}}$ with $|\tilde{b}_{\text{off}}| = \lceil \xi B \rceil$.

14:         Shape rewards for selected offline transitions: for each $(s, a, r, s') \in \tilde{b}_{\text{off}}$ set $\tilde{r} \leftarrow r + \beta \widehat{\Delta}(s, a)$.

15:         *# Critic update (use online batch + selected offline batch)*

16:         Define critic batch $b_Q \leftarrow b_{\text{on}} \cup \tilde{b}_{\text{off}}$ with rewards $r$ for online and $\tilde{r}$ for selected offline.

17:         **for** $i = 1, 2$ **do**

18:             Compute Bellman targets for $(s, a, r, s') \in b_Q$:

$$y(s, a, r, s') \;=\; r + \gamma\, \mathbb{E}_{a' \sim \pi_\varphi(\cdot \mid s')} \Big[ \min_{j=1,2} Q_{\varsigma_j}^{\text{tgt}}(s', a') - \alpha \log \pi_\varphi(a' \mid s') \Big].$$

19:             Update critic by gradient descent:

$$\varsigma_i \leftarrow \varsigma_i - \text{lr}\, \nabla_{\varsigma_i} \left( \frac{1}{|b_Q|} \sum_{(s,a,r,s') \in b_Q} \big(Q_{\varsigma_i}(s, a) - y(s, a, r, s')\big)^2 \right).$$

20:         **end for**

21:         *# Soft update target critics*

22:         **for** $i = 1, 2$ **do**

23:             $\varsigma_i^{\text{tgt}} \leftarrow \varpi\, \varsigma_i + (1 - \varpi)\, \varsigma_i^{\text{tgt}}.$

24:         **end for**

25:         *# Actor update (policy improvement + behavior cloning)*

26:         Define actor state batch $b_\pi \leftarrow \{s : (s, \cdot, \cdot, \cdot) \in b_{\text{on}} \cup \tilde{b}_{\text{off}}\}$.

27:         Sample $\tilde{a} \sim \pi_\varphi(\cdot \mid s)$ for each $s \in b_\pi$.

28:         Set normalization weight

$$\lambda \;=\; \frac{\omega}{\frac{1}{|b_\pi|} \sum_{s \in b_\pi} |\min_{i=1,2} Q_{\varsigma_i}(s, \tilde{a})| + \epsilon}, \qquad \epsilon > 0.$$

29:         Minimize actor loss:

$$\mathcal{L}_\pi(\varphi) \;=\; \frac{1}{|b_\pi|} \sum_{s \in b_\pi} \Big( \alpha \log \pi_\varphi(\tilde{a} \mid s) - \min_{i=1,2} Q_{\varsigma_i}(s, \tilde{a}) \Big) \;+\; \lambda \frac{1}{|b_{\text{off}}|} \sum_{(s,a,\cdot,\cdot) \in b_{\text{off}}} \| a - \bar{a} \|_2^2,$$

        where $\tilde{a} \sim \pi_\varphi(\cdot \mid s)$ for $s \in b_\pi$, and $\bar{a} \sim \pi_\varphi(\cdot \mid s)$ for $(s, a) \in b_{\text{off}}$.

30:         $\varphi \leftarrow \varphi - \text{lr}\, \nabla_\varphi \mathcal{L}_\pi(\varphi).$

31:     **end for**

32: **end for**

33: **Output:** Learned policy $\pi_\varphi$.

---

# F Proof of Lemma 2.2

**Lemma 2.2** (Return Bound between Two Environments [37]). *Let the empirical behavior policy in $\mathcal{D}_{\mathrm{off}}$ be $\pi_{\mathcal{D}_{\mathrm{off}}}(a \mid s)$. Define $C_1 = \frac{2r_{\max}}{(1-\gamma)^2}$. Then for any policy $\pi$,*

$$\eta_{\mathcal{M}_{\mathrm{on}}}(\pi) - \eta_{\mathcal{M}_{\mathrm{off}}}(\pi) \geq -2C_1 \mathbb{E}_{(s,a)\sim\rho_{\mathcal{M}}^{\pi_{\mathcal{D}_{\mathrm{off}}}}, s'\sim p_{\mathrm{off}}} \left[ D_{\mathrm{TV}}(\pi(\cdot|s') \,\|\, \pi_{\mathcal{D}_{\mathrm{off}}}(\cdot|s')) \right]$$
$$- C_1 \mathbb{E}_{(s,a)\sim\rho_{\mathcal{M}}^{\pi_{\mathcal{D}_{\mathrm{off}}}}} \left[ D_{\mathrm{TV}}(p_{\mathrm{on}}(\cdot|s,a) \,\|\, p_{\mathrm{off}}(\cdot|s,a)) \right].$$

*Proof.* Proof of Lemma 2.2 is already provided as an intermediate step for Theorem A.4 in [37], the offline performance bound case. We include their proof here with slight modifications for completeness.

We first cite the following two necessary lemmas also used in [37]. $\qquad\square$

**Lemma F.1** (Extended telescoping lemma). *Denote $\mathcal{M}_1 = (\mathcal{S}, \mathcal{A}, P_1, r, \gamma)$ and $\mathcal{M}_2 = (\mathcal{S}, \mathcal{A}, P_2, r, \gamma)$ as two MDPs that only differ in their transition dynamics. Suppose we have two policies $\pi_1, \pi_2$, we can reach the following conclusion:*

$$\eta_{\mathcal{M}_1}(\pi_1) - \eta_{\mathcal{M}_2}(\pi_2) = \frac{1}{1-\gamma} \mathbb{E}_{\rho_{\mathcal{M}_1}^{\pi_1}(s,a)} \left[ \mathbb{E}_{s'\sim P_1, a'\sim\pi_1} \left[ Q_{\mathcal{M}_2}^{\pi_2}(s',a') \right] - \mathbb{E}_{s'\sim P_2, a'\sim\pi_2} \left[ Q_{\mathcal{M}_2}^{\pi_2}(s',a') \right] \right].$$

*Proof.* This is Lemma C.2 in [65]. $\qquad\square$

**Lemma F.2.** *Denote $\mathcal{M} = (\mathcal{S}, \mathcal{A}, P, r, \gamma)$ as the underlying MDP. Suppose we have two policies $\pi_1, \pi_2$, then the performance difference of these policies in the MDP gives:*

$$\eta_{\mathcal{M}}(\pi_1) - \eta_{\mathcal{M}}(\pi_2) = \frac{1}{1-\gamma} \mathbb{E}_{\rho_{\mathcal{M}}^{\pi_1}(s,a), s'\sim P} \left[ \mathbb{E}_{a'\sim\pi_1} \left[ Q_{\mathcal{M}}^{\pi_2}(s',a') \right] - \mathbb{E}_{a'\sim\pi_2} \left[ Q_{\mathcal{M}}^{\pi_2}(s',a') \right] \right].$$

*Proof.* This is Lemma B.3 in [37]. $\qquad\square$

*Proof.* We have access to the offline data $\mathcal{M}_{\mathrm{off}}$ and the empirical behavioral policy $\pi_{\mathcal{D}_{\mathrm{off}}}$, so we bound the performance between $\eta_{\mathcal{M}_{\mathrm{on}}}(\pi)$ and $\eta_{\mathcal{M}_{\mathrm{off}}}(\pi_{\mathcal{D}_{\mathrm{off}}})$. We have

$$\eta_{\mathcal{M}_{\mathrm{on}}}(\pi) - \eta_{\pi_{\mathcal{D}_{\mathrm{off}}}}(\pi) = \underbrace{\eta_{\mathcal{M}_{\mathrm{on}}}(\pi) - \eta_{\mathcal{M}_{\mathrm{off}}}(\pi_{\mathcal{D}_{\mathrm{off}}})}_{(a)} + \underbrace{\eta_{\pi_{\mathcal{D}_{\mathrm{off}}}}(\pi_{\mathcal{D}_{\mathrm{off}}}) - \eta_{\pi_{\mathcal{D}_{\mathrm{off}}}}(\pi)}_{(b)}. \qquad (10)$$

To bound (a) term, we use Lemma F.1 in the second equality

$$\eta_{\mathcal{M}_{\mathrm{on}}}(\pi) - \eta_{\mathcal{M}_{\mathrm{off}}}(\mathcal{D}_{\mathrm{off}})$$
$$= -(\eta_{\mathcal{M}_{\mathrm{off}}}(\mathcal{D}_{\mathrm{off}}) - \eta_{\mathcal{M}_{\mathrm{on}}}(\pi))$$
$$= -\frac{1}{1-\gamma} \mathbb{E}_{(s,a)\sim\rho_{\mathcal{M}_{\mathrm{off}}}^{\mathcal{D}_{\mathrm{off}}}} \left[ \mathbb{E}_{s'_{\mathrm{off}}\sim p_{\mathcal{M}_{\mathrm{off}}}, a'\sim\pi_{\mathcal{D}_{\mathrm{off}}}} \left[ Q_{\mathcal{M}_{\mathrm{on}}}^{\pi}(s'_{\mathrm{off}}, a') \right] - \mathbb{E}_{s'_{\mathrm{on}}\sim p_{\mathcal{M}_{\mathrm{on}}}, a'\sim\pi} \left[ Q_{\mathcal{M}_{\mathrm{on}}}^{\pi}(s'_{\mathrm{on}}, a') \right] \right]$$
$$= -\frac{1}{1-\gamma} \mathbb{E}_{(s,a)\sim\rho_{\mathcal{M}_{\mathrm{off}}}^{\mathcal{D}_{\mathrm{off}}}} \Big[$$
$$\underbrace{\left( \mathbb{E}_{s'_{\mathrm{off}}\sim p_{\mathcal{M}_{\mathrm{off}}}, a'\sim\pi_{\mathcal{D}_{\mathrm{off}}}} \left[ Q_{\mathcal{M}_{\mathrm{on}}}^{\pi}(s'_{\mathrm{off}}, a') \right] - \mathbb{E}_{s'_{\mathrm{off}}\sim p_{\mathcal{M}_{\mathrm{off}}}, a'\sim\pi} \left[ Q_{\mathcal{M}_{\mathrm{on}}}^{\pi}(s'_{\mathrm{off}}, a') \right] \right)}_{(c)}$$
$$+ \underbrace{\left( \mathbb{E}_{s'_{\mathrm{off}}\sim p_{\mathcal{M}_{\mathrm{off}}}, a'\sim\pi} \left[ Q_{\mathcal{M}_{\mathrm{on}}}^{\pi}(s'_{\mathrm{off}}, a') \right] - \mathbb{E}_{s'_{\mathrm{on}}\sim p_{\mathcal{M}_{\mathrm{on}}}, a'\sim\pi} \left[ Q_{\mathcal{M}_{\mathrm{on}}}^{\pi}(s'_{\mathrm{on}}, a') \right] \right)}_{(d)} \Big].$$

To bound term (c), we use

$$\mathbb{E}_{s'_{\mathrm{off}}\sim p_{\mathcal{M}_{\mathrm{off}}}, a'\sim\pi_{\mathcal{D}_{\mathrm{off}}}} \left[ Q_{\mathcal{M}_{\mathrm{on}}}^{\pi}(s'_{\mathrm{off}}, a') \right] - \mathbb{E}_{s'_{\mathrm{off}}\sim p_{\mathcal{M}_{\mathrm{off}}}, a'\sim\pi} \left[ Q_{\mathcal{M}_{\mathrm{on}}}^{\pi}(s'_{\mathrm{off}}, a') \right]$$
$$\leq \mathbb{E}_{s'_{\mathrm{off}}\sim p_{\mathcal{M}_{\mathrm{off}}}} \left[ \sum_{a'\in\mathcal{A}} |\mathcal{D}_{\mathrm{off}}(a' \mid s'_{\mathrm{off}}) - \pi(a' \mid s'_{\mathrm{off}})| \cdot \left| Q_{\mathcal{M}_{\mathrm{on}}}^{\pi}(s'_{\mathrm{off}}, a') \right| \right]$$
$$\leq \frac{2r_{\max}}{1-\gamma} \mathbb{E}_{s'_{\mathrm{off}}\sim p_{\mathcal{M}_{\mathrm{off}}}} \left[ D_{\mathrm{TV}}\left( \pi_{\mathcal{D}_{\mathrm{off}}}(\cdot \mid s'_{\mathrm{off}}) \,\|\, \pi(\cdot \mid s'_{\mathrm{off}}) \right) \right],$$

where the last inequality comes from the fact that $|Q^\pi_{\mathcal{M}_{\text{on}}}(s'_{\text{off}}, a')| \leq \frac{r_{\max}}{1-\gamma}$ and the definition of TV distance.

$$
\begin{aligned}
(d) &= \mathbb{E}_{s' \sim p_{\mathcal{M}_{\text{off}}}, \, a' \sim \pi} \left[ Q^\pi_{\mathcal{M}_{\text{on}}}(s', a') \right] - \mathbb{E}_{s' \sim p_{\mathcal{M}_{\text{on}}}, \, a' \sim \pi} \left[ Q^\pi_{\mathcal{M}_{\text{on}}}(s', a') \right] \\
&= \int_{\mathcal{S}} \left( p_{\mathcal{M}_{\text{off}}}(s' \mid s, a) - p_{\mathcal{M}_{\text{on}}}(s' \mid s, a) \right) \left( \sum_{a'} \pi(a' \mid s') Q^\pi_{\mathcal{M}_{\text{on}}}(s', a') \right) ds' \\
&\leq \frac{r_{\max}}{1-\gamma} \int_{\mathcal{S}} \left| p_{\mathcal{M}_{\text{off}}}(s' \mid s, a) - p_{\mathcal{M}_{\text{on}}}(s' \mid s, a) \right| ds' \\
&= \frac{2 r_{\max}}{1-\gamma} \left[ D_{\text{TV}} \left( p_{\mathcal{M}_{\text{off}}}(\cdot \mid s, a) \,\|\, p_{\mathcal{M}_{\text{on}}}(\cdot \mid s, a) \right) \right],
\end{aligned}
$$

where the inequality comes from the fact that $Q^\pi_{\mathcal{M}_{\text{on}}}(s', a')$ weighted by probability is bounded by $\frac{r_{\max}}{1-\gamma}$, the upper bound for all Q-values.

Combine the bounds for (c) and (d), we obtain a bound for the (a) term:

$$
\begin{aligned}
\eta_{\mathcal{M}_{\text{on}}}(\pi) - \eta_{\mathcal{M}_{\text{off}}}(\pi_{\mathcal{D}_{\text{off}}}) \geq & -\frac{2 r_{\max}}{(1-\gamma)^2} \mathbb{E}_{(s,a) \sim \rho^{\pi_{\mathcal{D}_{\text{off}}}}_{\mathcal{M}_{\text{off}}}, \, s' \sim p_{\mathcal{M}_{\text{off}}}} \left[ D_{\text{TV}} \left( \pi_{\mathcal{D}_{\text{off}}}(\cdot \mid s') \,\|\, \pi(\cdot \mid s') \right) \right] \\
& -\frac{2 r_{\max}}{(1-\gamma)^2} \mathbb{E}_{(s,a) \sim \rho^{\pi_{\mathcal{D}_{\text{off}}}}_{\mathcal{M}_{\text{off}}}} \left[ D_{\text{TV}} \left( p_{\mathcal{M}_{\text{off}}}(\cdot \mid s, a) \,\|\, p_{\mathcal{M}_{\text{on}}}(\cdot \mid s, a) \right) \right].
\end{aligned}
$$

Now we try to bound term (b).

$$
\begin{aligned}
& \eta_{\mathcal{M}_{\text{off}}}(\pi_{\mathcal{D}_{\text{off}}}) - \eta_{\mathcal{M}_{\text{off}}}(\pi) \\
&= \frac{1}{1-\gamma} \mathbb{E}_{(s,a) \sim \rho^{\pi_{\mathcal{D}_{\text{off}}}}_{\mathcal{M}_{\text{off}}}, \, s' \sim p_{\mathcal{M}_{\text{off}}}(\cdot \mid s, a)} \left[ \mathbb{E}_{a' \sim \pi_{\mathcal{D}_{\text{off}}}} \left[ Q^\pi_{\mathcal{M}_{\text{off}}}(s', a') \right] - \mathbb{E}_{a' \sim \pi} \left[ Q^\pi_{\mathcal{M}_{\text{off}}}(s', a') \right] \right] \\
&\geq -\frac{1}{1-\gamma} \mathbb{E}_{(s,a) \sim \rho^{\pi_{\mathcal{D}_{\text{off}}}}_{\mathcal{M}_{\text{off}}}, \, s' \sim p_{\mathcal{M}_{\text{off}}}(\cdot \mid s, a)} \left| \mathbb{E}_{a' \sim \pi_{\mathcal{D}_{\text{off}}}} \left[ Q^\pi_{\mathcal{M}_{\text{off}}}(s', a') \right] - \mathbb{E}_{a' \sim \pi} \left[ Q^\pi_{\mathcal{M}_{\text{off}}}(s', a') \right] \right| \\
&\geq -\frac{1}{1-\gamma} \mathbb{E}_{(s,a) \sim \rho^{\pi_{\mathcal{D}_{\text{off}}}}_{\mathcal{M}_{\text{off}}}, \, s' \sim p_{\mathcal{M}_{\text{off}}}(\cdot \mid s, a)} \left| \sum_{a' \in \mathcal{A}} \left( \pi_{\mathcal{D}_{\text{off}}}(a' \mid s') - \pi(a' \mid s') \right) Q^\pi_{\mathcal{M}_{\text{off}}}(s', a') \right| \\
&\geq -\frac{r_{\max}}{(1-\gamma)^2} \mathbb{E}_{(s,a) \sim \rho^{\pi_{\mathcal{D}_{\text{off}}}}_{\mathcal{M}_{\text{off}}}, \, s' \sim p_{\mathcal{M}_{\text{off}}}(\cdot \mid s, a)} \left| \sum_{a' \in \mathcal{A}} \left( \pi_{\mathcal{D}_{\text{off}}}(a' \mid s') - \pi(a' \mid s') \right) \right| \\
&= -\frac{2 r_{\max}}{(1-\gamma)^2} \mathbb{E}_{(s,a) \sim \rho^{\pi_{\mathcal{D}_{\text{off}}}}_{\mathcal{M}_{\text{off}}}, \, s' \sim p_{\mathcal{M}_{\text{off}}}(\cdot \mid s, a)} \left[ D_{\text{TV}} \left( \pi_{\mathcal{D}_{\text{off}}}(\cdot \mid s') \,\|\, \pi(\cdot \mid s') \right) \right],
\end{aligned}
$$

where the first equality is a direct application of Lemma F.2. Combine the bound for term (a) and (b), we conclude that

$$
\begin{aligned}
\eta_{\mathcal{M}_{\text{on}}}(\pi) - \eta_{\mathcal{M}_{\text{off}}}(\pi) \geq & -2 C_1 \mathbb{E}_{(s,a) \sim \rho^{\pi_{\mathcal{D}_{\text{off}}}}_{\mathcal{M}}, \, s' \sim p_{\text{off}}} \left[ D_{\text{TV}} \left( \pi(\cdot \mid s') \,\|\, \pi_{\mathcal{D}_{\text{off}}}(\cdot \mid s') \right) \right] \\
& - C_1 \mathbb{E}_{(s,a) \sim \rho^{\pi_{\mathcal{D}_{\text{off}}}}_{\mathcal{M}}} \left[ D_{\text{TV}} \left( p_{\text{on}}(\cdot \mid s, a) \,\|\, p_{\text{off}}(\cdot \mid s, a) \right) \right],
\end{aligned}
$$

where $C_1 = \frac{2 r_{\max}}{(1-\gamma)^2}$. $\qquad\square$

## G  Proof of Theorem 3.1

**Definition G.1.** For any distributions $p_0, p_1$ on $R^d$ with finite second moments, define $D_2(p_0, p_1) := E\|X_1 - X_0\|_2^2$, $X_0 \sim p_0$, $X_1 \sim p_1$ independent.

**Theorem 3.1** (Conditions for Composite Flow Yielding Smaller Errors). *Assume that composite flow and direct flow share the same hypothesis class $\mathcal{H}$ of (measurable) vector fields $v : \mathbb{R}^d \times [0,1] \to \mathbb{R}^d$, and that there exists $B \in (0, \infty)$ such that $\sup_{v \in \mathcal{H}} \sup_{x \in \mathbb{R}^d, \, t \in [0,1]} \|v(x, t)\|_2 \leq B$. Also, assume that $\max\{\text{Tr}(\Sigma_{\text{on}}), \text{Tr}(\widehat{\Sigma}_{\text{off}})\} \leq C_{\text{TR}}$ for some $C_{\text{TR}}$. The composite flow enjoys a strictly tighter high-probability generalization bound than the direct flow if and only if*

$$
W_2(p_G, p_{\text{on}}) > W_2(\hat{p}_{\text{off}}, p_{\text{on}}) \tag{2}
$$

*Here, $p_G := \mathcal{N}(0, I_d)$, $\Sigma_{\text{on}}$ is the covariance of $p_{\text{on}}$, and $\widehat{\Sigma}_{\text{off}}$ is the covariance of $\hat{p}_{\text{off}}$.*

*Proof.* We consider *linear-path* flow matching (FM) trained with squared loss. A single training example is generated by: $t \sim \text{Unif}[0, 1]$; $X_0 \sim p_0$ (the *start* distribution), independently of $t$; $X_1 \sim p_{\text{on}}$, independently of $(t, X_0)$; The interpolation $X_t := (1 - t)X_0 + tX_1$ and the label $Y := X_1 - X_0 \in \mathbb{R}^d$.

Let $\mathcal{H}$ be a hypothesis class of (measurable) vector fields $v : \mathbb{R}^d \times [0, 1] \to \mathbb{R}^d$ and define the population risk

$$R(v) := \mathbb{E}\Big[ \|v(X_t, t) - Y\|_2^2 \Big], \tag{11}$$

where the expectation is over the sampling mechanism above. Let

$$v^\star(x, t) := \mathbb{E}[Y \mid X_t = x, \, t] \tag{12}$$

denote the Bayes (population) minimizer. Given $n$ i.i.d. samples, let $\hat{v}$ be any empirical risk minimizer over $\mathcal{H}$ for the squared loss.

**Assumptions.** **A1** (*Uniform boundedness*) There exists $B \in (0, \infty)$ such that $\sup_{v \in \mathcal{H}} \sup_{x \in \mathbb{R}^d, \, t \in [0,1]} \|v(x, t)\|_2 \le B$. This ensures integrability and controls the Lipschitz constants used below.

**FM as regression; Bayes predictor and excess-risk identity.** By definition,

$$R(v) = \mathbb{E} \, \|v(X_t, t) - Y\|_2^2, \quad Y = X_1 - X_0.$$

For any measurable $v$, the *Bayes* (population) minimizer for the squared loss is the conditional mean

$$v^\star(x, t) = \mathbb{E}[Y \mid X_t = x, \, t].$$

A standard identity for squared loss (we derive it fully) is

$$R(v) - R(v^\star) = \mathbb{E} \, \| v(X_t, t) - v^\star(X_t, t) \|_2^2. \tag{13}$$

*Derivation of* (13). Expand and add/subtract $v^\star(X_t, t)$:

$$R(v) = \mathbb{E} \, \| v(X_t, t) - v^\star(X_t, t) + v^\star(X_t, t) - Y \|_2^2$$
$$= \mathbb{E} \, \|v - v^\star\|_2^2 + 2 \mathbb{E} \langle v - v^\star, \, v^\star - Y \rangle + \mathbb{E} \, \|v^\star - Y\|_2^2,$$

where all functions are evaluated at $(X_t, t)$ but notationally suppressed for readability. Condition on $(X_t, t)$: by definition, $\mathbb{E}[Y \mid X_t, t] = v^\star(X_t, t)$, hence $\mathbb{E}[v^\star - Y \mid X_t, t] = 0$ and the cross term vanishes after taking expectations. Therefore

$$R(v) = \mathbb{E} \, \|v - v^\star\|_2^2 + R(v^\star),$$

which rearranges to (13).

In Lemma G.3, we prove that there exists an absolute constant $c > 0$ such that, for all $\delta \in (0, 1)$, with probability at least $1 - \delta$ (over the sample draw),

$$R(\hat{v}) \le R(v^\star) + c \left( B + \sqrt{\mathbb{E} \, \|Y\|_2^2} \right) \Gamma_{n,\delta}, \qquad \Gamma_{n,\delta} := \mathfrak{R}_n(\mathcal{H}) + \sqrt{\tfrac{\log(1/\delta)}{n}}, \tag{14}$$

where $\mathfrak{R}_n(\mathcal{H})$ is the (data-independent) Rademacher complexity of $\mathcal{H}$. In short: symmetrization + vector contraction + Lipschitzness of the squared loss on the $B$-ball yields (14).)

Subtract $R(v^\star)$ on both sides of (14) and apply (13) with $v = \hat{v}$:

$$\mathbb{E} \, \|\hat{v} - v^\star\|_2^2 \le c \left( B + \sqrt{\mathbb{E} \, \|Y\|_2^2} \right) \Gamma_{n,\delta}. \tag{15}$$

**Identify $\mathbb{E} \, \|Y\|_2^2$ as $D_2(p_0, p_{\text{on}})$ and linearize the square root.** By independence of $X_0 \sim p_0$ and $X_1 \sim p_{\text{on}}$,

$$\mathbb{E} \, \|Y\|_2^2 = \mathbb{E} \, \|X_1 - X_0\|_2^2 =: D_2(p_0, p_{\text{on}}). \tag{16}$$

By applying Lemma G.4, we have

$$\sqrt{\mathbb{E} \, \|Y\|_2^2} \le \sqrt{\text{Tr}(\Sigma_{\text{on}}) + C_{\text{TR}}} + W_2(p_0, p_{\text{on}}). \tag{17}$$

Therefore, we obtain

$$\mathbb{E} \, \|\hat{v} - v^\star\|_2^2 \leq c \left( B + \sqrt{\mathrm{Tr}(\Sigma_{\mathrm{on}}) + C_{\mathrm{TR}}} + W_2(p_0, p_{\mathrm{on}}) \right) \Gamma_{n,\delta} \tag{18}$$

This gives the following master bound (19) with $C_0 := c(B + \sqrt{\mathrm{Tr}(\Sigma_{\mathrm{on}}) + C_{\mathrm{TR}}}$ and $C_1 := c$.

**Master bound.** Under **A1**, for any start $p_0$ (with fixed target $p_{\mathrm{on}}$), with probability at least $1 - \delta$,

$$\mathbb{E} \, \|\hat{v} - v^\star\|_2^2 \, \leq \, \underbrace{C_0 \, \Gamma_{n,\delta}}_{p_0\text{-independent}} \, + \, \underbrace{C_1 \, W_2(p_0, p_{\mathrm{on}}) \, \Gamma_{n,\delta}}_{\text{depends on the start } p_0}, \qquad C_0 := c(B + \sqrt{\mathrm{Tr}(\Sigma_{\mathrm{on}}) + C_{\mathrm{TR}}}), \ \ C_1 := c.$$

$$\tag{19}$$

**Composite vs. direct.** Consider two trainings sharing the same online $p_{\mathrm{on}}$ and class $\mathcal{H}$:

$$\textbf{(Composite step)} \quad p_0 \; = \; \hat{p}_{\mathrm{off}}(\cdot \mid s, a), \tag{20}$$
$$\textbf{(Direct step)} \quad p_0 \; = \; p_G \; = \; \mathcal{N}(0, I_d). \tag{21}$$

Applying (19) to (20) and (21) yields

$$\mathbb{E} \, \left\| \hat{v}_{\mathrm{comp}} - v^\star_{\mathrm{comp}} \right\|_2^2 \; \leq \; C_0 \, \Gamma_{n,\delta} + C_1 \, W_2\big(\hat{p}_{\mathrm{off}}, p_{\mathrm{on}}\big) \, \Gamma_{n,\delta}, \tag{22}$$
$$\mathbb{E} \, \|\hat{v}_G - v^\star_G\|_2^2 \; \leq \; C_0 \, \Gamma_{n,\delta} + C_1 \, W_2\big(p_G, p_{\mathrm{on}}\big) \, \Gamma_{n,\delta}. \tag{23}$$

**When is the composite bound strictly tighter?** Applying (19) to $p_0 = \hat{p}_{\mathrm{off}}$ (composite) and $p_0 = p_G$ (direct) gives (22) and (23). Since $C_0$ and $\Gamma_{n,\delta}$ are identical across the two trainings (they depend on $(B, c, \Sigma_{\mathrm{on}}, C_{\mathrm{TR}})$, $n$, $\delta$, and $\mathcal{H}$, but not on $p_0$), the *only* difference in the right-hand sides is the factor $W_2(\cdot, p_{\mathrm{on}})$.

Therefore the composite bound (22) is *strictly smaller* than the direct bound (23) if and only if

$$W_2\big(\hat{p}_{\mathrm{off}}, p_{\mathrm{on}}\big) \; < \; W_2\big(p_G, p_{\mathrm{on}}\big). \tag{24}$$

$\square$

*Remark* G.2 (Optional $D_2$ strengthening). One can analogously replace $W_2(p_0, p_{\mathrm{on}})$ by $D_2(p_0, p_{\mathrm{on}})$ when bounding 15, using the standard inequality trick $\sqrt{x} \leq \frac{1}{2}(x + 1)$ for all $x \geq 0$. We use $W_2$ metric because it's more commonly adopted in the literature, but $D_2$ leads to a tighter bound since we no longer needs the constant $C_{\mathrm{TR}}$. We do not expand the $D_2$ case here, as the argument mirrors the $W_2$ case.

**Lemma G.3** (Generalization gap for squared loss under uniform boundedness). *Assume A1. Let* $\ell_v(z) := \|v(x, t) - y\|_2^2$ *for* $z = (x, t, y)$ *generated as in the theorem. Then there exists a positive constant* $c > 0$ *such that, for any* $\delta \in (0, 1)$, *with probability at least* $1 - \delta$ *over an i.i.d. sample of size* $n$,

$$R(\hat{v}) \; \leq \; R(v^\star) \; + \; c \left( B + \sqrt{\mathbb{E} \, \|Y\|_2^2} \right) \Gamma_{n,\delta}, \qquad \Gamma_{n,\delta} := \mathfrak{R}_n(\mathcal{H}) + \sqrt{\tfrac{\log(1/\delta)}{n}}.$$

Proof. *We sketch the standard steps and make constants explicit where needed:*

(i) Symmetrization. *Let* $\hat{R}(v)$ *denote the empirical risk. For ERM* $\hat{v}$,

$$R(\hat{v}) - R(v^\star) \; \leq \; 2 \sup_{v \in \mathcal{H}} \big( R(v) - \hat{R}(v) \big)$$

*up to negligible terms; this is standard (e.g., by a one-sided symmetrization plus a chaining step). Thus it suffices to bound* $\sup_{v \in \mathcal{H}} (R(v) - \hat{R}(v))$.

(ii) Lipschitz envelope via contraction. *For fixed* $(x, t, y)$ *with* $\|v(x, t)\| \leq B$ *and any* $u \in \mathbb{R}^d$ *with* $\|u\| \leq B$,

$$\Big| \, \|u - y\|^2 - \|v(x, t) - y\|^2 \, \Big| = \big| \, \langle u - v(x, t), \, (u + v(x, t) - 2y) \rangle \, \big| \leq 2 \, (B + \|y\|) \, \|u - v(x, t)\|.$$

*Thus, as a function of $u$, the map $u \mapsto \|u - y\|^2$ is $(2(B + \|y\|))$-Lipschitz on the B-ball. Applying the vector contraction inequality to the class $\{v(\cdot, \cdot) \in \mathcal{H}\}$ gives*

$$\mathbb{E} \sup_{v \in \mathcal{H}} \big(R(v) - \hat{R}(v)\big) \lesssim \Big(\mathbb{E}\,[\,B + \|Y\|\,]\Big)\mathfrak{R}_n(\mathcal{H}).$$

*By Cauchy–Schwarz and Jensen, $\mathbb{E}\,\|Y\| \leq \sqrt{\mathbb{E}\,\|Y\|^2}$.*

(iii) High-probability upgrade. *A standard bounded differences (or Bernstein-type) argument for Lipschitz, sub-quadratic losses upgrades the expected sup-gap to a high-probability bound, adding the usual $\sqrt{\log(1/\delta)/n}$ term. Collecting constants and using $\mathbb{E}\,\|Y\| \leq \sqrt{\mathbb{E}\,\|Y\|^2}$ yields the claimed form with some absolute $c > 0$:*

$$\sup_{v \in \mathcal{H}} \big(R(v) - \hat{R}(v)\big) \leq c\Big(B + \sqrt{\mathbb{E}\,\|Y\|^2}\Big)\Gamma_{n,\delta}.$$

*Since $\hat{v}$ is an ERM, $\hat{R}(\hat{v}) \leq \hat{R}(v^\star)$, and therefore $R(\hat{v}) - R(v^\star) \leq 2\sup_v(R(v) - \hat{R}(v))$ in the final step, absorbing the factor 2 into $c$.* $\qquad\square$

**Lemma G.4.** *Let $\mu.$ and $\Sigma.$ denote mean and covariance, respectively. We show that if $\mathrm{Tr}(\Sigma_0) \leq C_{\mathrm{TR}}$ for some $C_{\mathrm{TR}} > 0$, we have:*

$$\sqrt{D_2(p_0, p_1)} \leq \sqrt{\mathrm{Tr}(\Sigma_1) + C_{\mathrm{TR}}} + W_2(p_0, p_1)$$

*Proof.* Let $X_0 \sim p_0$ and $X_1 \sim p_1$ be independent with means $\mu_0, \mu_1$ and covariances $\Sigma_0, \Sigma_1$. We compute

$$D_2(p_0, p_1) = \mathbb{E}\,\|X_1 - X_0\|_2^2 = \mathbb{E}\,\|X_1\|_2^2 + \mathbb{E}\,\|X_0\|_2^2 - 2\,\mathbb{E}\,\langle X_1, X_0 \rangle.$$

By independence, $\mathbb{E}\,\langle X_1, X_0 \rangle = \langle \mathbb{E}X_1, \mathbb{E}X_0 \rangle = \langle \mu_1, \mu_0 \rangle$. Moreover, for any random vector $Z$ with mean $\mu$ and covariance $\Sigma$, $\mathbb{E}\,\|Z\|_2^2 = \|\mu\|_2^2 + \mathrm{Tr}(\Sigma)$. Thus

$$\begin{aligned}
D_2(p_0, p_1) &= \big(\|\mu_1\|_2^2 + \mathrm{Tr}(\Sigma_1)\big) + \big(\|\mu_0\|_2^2 + \mathrm{Tr}(\Sigma_0)\big) - 2\langle \mu_1, \mu_0 \rangle \\
&= \|\mu_1 - \mu_0\|_2^2 + \mathrm{Tr}(\Sigma_1 + \Sigma_0),
\end{aligned}$$

Let $\Gamma(p_0, p_1)$ be all couplings of $p_0, p_1$. For any $\pi \in \Gamma(p_0, p_1)$ with $(X_0, X_1) \sim \pi$, we have

$$\mathbb{E}_\pi \|X_1 - X_0\|_2^2 \geq \big\|\mathbb{E}_\pi[X_1 - X_0]\big\|_2^2 = \|\mu_1 - \mu_0\|_2^2,$$

by Jensen's inequality since $z \mapsto \|z\|_2^2$ is convex. Taking the infimum over all couplings gives

$$W_2^2(p_0, p_1) = \inf_{\pi \in \Gamma(p_0, p_1)} \mathbb{E}_\pi \|X_1 - X_0\|_2^2 \geq \|\mu_1 - \mu_0\|_2^2.$$

Therefore, we have $\|\mu_1 - \mu_0\|_2^2 \leq W_2^2(p_0, p_1)$. Hence,

$$\begin{aligned}
\sqrt{D_2(p_0, p_1)} &= \sqrt{\|\mu_1 - \mu_0\|_2^2 + \mathrm{Tr}(\Sigma_1 + \Sigma_0)} \\
&\leq \sqrt{\mathrm{Tr}(\Sigma_0 + \Sigma_1)} + \sqrt{\|\mu_1 - \mu_0\|_2^2} \\
&\leq \sqrt{\mathrm{Tr}(\Sigma_1) + C_{\mathrm{TR}}} + W_2(p_0, p_1)
\end{aligned}$$

This completes the proof. $\qquad\square$

The assumption that $\mathrm{Tr}(\Sigma_0) \leq C_{\mathrm{TR}}$ makes sense because we assume bounded state space for all settings considered in our paper.

# H  Proof of Theorem 3.5

**Theorem 3.5** (Large Dynamics Gap Exploration Reduces Performance Gap).  *Compared to behavior cloning policy $\pi_{\mathrm{bc}}$ on the offline dataset, training a policy $\hat{\pi}$ by replacing all offline samples with a dynamics gap exceeding $\kappa$ (as estimated by the composite flow) with online environment samples can reduce the performance gap to the optimal online policy $\pi_{on}^*$ with high probability by*

$$\frac{2L_r(1+\gamma)}{(1-\gamma)(1-\gamma L_p)}\left(\Delta_{W_2} - \kappa - \sqrt{(C_0 + C_1\, W_2(\hat{p}_{\mathrm{off}}, p_{\mathrm{on}}))\,\Gamma_{N_{\mathrm{on}},\delta}}\right). \tag{7}$$

*Here $L_r$ and $L_P$ are the Lipschitz constants for the reward and transition function, respectively. $\gamma L_p < 1$. $\Delta_{W_2} := \sup_{s,a} W_2\big(p_{\mathrm{off}}(\cdot|s,a),\, p_{\mathrm{on}}(\cdot|s,a)\big)$ is the largest dynamics gap. $C_0$ and $C_1$ are two constants. $\Gamma_{N_{\mathrm{on}},\delta} := \mathfrak{R}_{N_{\mathrm{on}}}(\mathcal{H}) + \sqrt{\frac{\log(1/\delta)}{N_{\mathrm{on}}}}$, where $\mathcal{H}$ is the same as in Theorem 3.1 and $N_{\mathrm{on}}$ is the number of samples used to train the online flow.*

*Proof.*  Let's first list the assumptions:

**R1: Lipschitz reward.** For every $a \in \mathcal{A}$, $s \mapsto r(s,a)$ is $L_r$-Lipschitz.

**R2: Lipschitz dynamics.** For $m \in \{\mathrm{off}, \mathrm{on}\}$, $W_1\big(p_m(\cdot \mid s,a), p_m(\cdot \mid s',a)\big) \le L_p\, d(s,s')$ for all $s, s', a$.

**R3: Contraction.** $\gamma L_p < 1$.

We proceed in four steps: (A) a policy-dependent model-gap bound; (B) a uniform high-probability $W_2$-gap estimation bound for the rectified flow ($\alpha$); (C) learning-gap bounds for $\pi_{\mathrm{bc}}$ and $\hat{\pi}$ ($\beta$); (D) assembly via the standard three-term decomposition.

**Step A: Policy-dependent model-gap bound.**  For any fixed policy $\pi$, we cite Theorem 3.1 of [45] for the following bound. Under **R1**–**R3**, we have

$$\begin{aligned}
\|V_{\mathrm{on}}^\pi - V_{\mathrm{off}}^\pi\|_\infty &\le 2L_r\Delta_{W_1}(1+\gamma)\sum_{i=0}^\infty \gamma^i \sum_{j=0}^i (L_P)^j \\
&= 2L_r\Delta_{W_1}(1+\gamma)\sum_{j=0}^\infty (L_P)^j \sum_{i=j}^\infty \gamma^i \\
&= 2L_r\Delta_{W_1}(1+\gamma)\sum_{j=0}^\infty (L_P)^j \cdot \frac{\gamma^j}{1-\gamma} \\
&= \frac{2L_r\Delta_{W_1}(1+\gamma)}{1-\gamma}\sum_{j=0}^\infty (\gamma L_P)^j \\
&= \frac{2L_r\Delta_{W_1}(1+\gamma)}{(1-\gamma)(1-\gamma L_P)}
\end{aligned} \tag{25}$$

Averaging over the initial state-distribution $\mu$ gives us a bound on what we call the model gap:

$$\begin{aligned}
\left|\eta_{\mathcal{M}_{\mathrm{on}}}(\pi) - \eta_{\mathcal{M}_{\mathrm{off}}}(\pi)\right| &= \left|\int_{\mathcal{S}}\big(V_{\mathrm{on}}^\pi(s) - V_{\mathrm{off}}^\pi(s)\big)\,\mu(ds)\right| \\
&\le \int_{\mathcal{S}}\left|V_{\mathrm{on}}^\pi(s) - V_{\mathrm{off}}^\pi(s)\right|\mu(ds). \\
&\le \|V_{\mathrm{on}}^\pi - V_{\mathrm{off}}^\pi\|_\infty \\
&\le \frac{2L_r\Delta_{W_1}(1+\gamma)}{(1-\gamma)(1-\gamma L_p)}.
\end{aligned}$$

Moreover, the bounded state space assumption in Theorem 3.1 implies bounded second moment for both $p_{\text{off}}$ and $p_{\text{on}}$, so we have $W_1(p_{\text{off}}, p_{\text{on}}) \leq W_2(p_{\text{off}}, p_{\text{on}}) \ \forall p_{\text{off}}, p_{\text{on}}$. Hence, we obtain that

$$|\eta_{\mathcal{M}_{\text{on}}}(\pi) - \eta_{\mathcal{M}_{\text{off}}}(\pi)| \ \leq \ \frac{2L_r(1+\gamma)}{(1-\gamma)(1-\gamma L_p)} \Delta_{W_1} \tag{26}$$

$$\leq \ \frac{2L_r(1+\gamma)}{(1-\gamma)(1-\gamma L_p)} \Delta_{W_2}. \tag{27}$$

**Step B: FM-based uniform $W_2$ estimation bound (construction of $\alpha$).** We need a high-probability *uniform* control of the error of the rectified-flow $W_2$ estimator to ensure that thresholding by $\kappa$ actually caps the *on-policy* mismatch for $\hat{\pi}$.

*B.1. Flow-to-map error.* Let $T$ be the Brenier OT map from $\hat{p}_{\text{off}}(\cdot \mid s, a)$ to $p_{\text{on}}(\cdot \mid s, a)$ (guaranteed by standard assumptions), and let $\widehat{T}$ be the terminal map obtained by integrating the learned FM velocity field $\hat{v}$ from $t = 0$ to $t = 1$. Under the linear path and Lipschitz regularity of $v^\star$ and $\hat{v}$ (Theorem 3.1 assumptions), a stability argument (Gronwall) yields

$$\mathbb{E}_{X_0 \sim \hat{p}_{\text{off}}} \big\| \widehat{T}(X_0) - T(X_0) \big\|_2^2 \ \leq \ K_{\text{stab}} \int_0^1 \mathbb{E} \big\| \hat{v}(X_t, t) - v^\star(X_t, t) \big\|_2^2 \, dt, \tag{28}$$

for some finite $K_{\text{stab}}$ depending on Lipschitz constants of the flow. By the "master bound" (Eq. (7) in Theorem 3.1), uniformly over $t \in [0, 1]$, with prob. $\geq 1 - \delta$,

$$\mathbb{E} \big\| \hat{v} - v^\star \big\|_2^2 \ \leq \ C_0 \, \Gamma_{N_{\text{on}}, \delta} + C_1 \, W_2(\hat{p}_{\text{off}}, p_{\text{on}}) \, \Gamma_{N_{\text{on}}, \delta}. \tag{29}$$

So we have

$$\underbrace{\mathbb{E} \| \widehat{T}(X_0) - T(X_0) \|_2^2}_{\text{terminal map MSE}} \overset{\text{Gronwall}}{\leq} K_{\text{stab}} \int_0^1 \underbrace{\mathbb{E} \| \hat{v} - v^\star \|_2^2}_{\text{FM excess risk at } t} \, dt \tag{30}$$

$$\leq K_{\text{stab}} \left( C_0 + C_1 \, W_2(\hat{p}_{\text{off}}, p_{\text{on}}) \right) \Gamma_{N_{\text{on}}, \delta} \tag{31}$$

*B.2. From map MSE to $W_2$ error.* By the coupling construction $(\hat{s}, s) = (\widehat{T}(X_0), T(X_0))$ with $X_0 \sim \hat{p}_{\text{off}}$ and the triangle inequality for $W_2$,

$$\sup_{(s,a)} \left| \widehat{\Delta}_{W_2}(s, a) - \Delta_{W_2}(s, a) \right| \ \leq \ \sup_{(s,a)} W_2\big(\hat{p}_{\text{on}}, p_{\text{on}}\big) \tag{32}$$

Define the coupling $\gamma_{(s,a)} := (T, \widehat{T})_{\#} \hat{p}_{\text{off}}(\cdot \mid s, a)$, i.e., if $S := T(X_0)$ and $\widehat{S} := \widehat{T}(X_0)$ then $(S, \widehat{S}) \sim \gamma_{(s,a)}$ and $\gamma_{(s,a)} \in \Pi\big(p_{\text{on}}(\cdot \mid s, a), \hat{p}_{\text{on}}(\cdot \mid s, a)\big)$. By the definition of the 2-Wasserstein distance,

$$W_2^2\big(p_{\text{on}}(\cdot \mid s, a), \hat{p}_{\text{on}}(\cdot \mid s, a)\big) = \inf_{\gamma \in \Pi(p_{\text{on}}, \hat{p}_{\text{on}})} \int \|x - y\|_2^2 \, d\gamma(x, y) \tag{33}$$

$$\leq \int \|x - y\|_2^2 \, d\gamma_{(s,a)}(x, y) = \mathbb{E} \|\widehat{T}(X_0) - T(X_0)\|_2^2. \tag{34}$$

Taking square roots gives, for each $(s, a)$,

$$W_2\big(p_{\text{on}}(\cdot \mid s, a), \hat{p}_{\text{on}}(\cdot \mid s, a)\big) \ \leq \ \sqrt{\mathbb{E} \|\widehat{T}(X_0) - T(X_0)\|_2^2}.$$

Finally, taking the supremum over $(s, a)$ preserves the inequality:

$$\sup_{(s,a)} W_2\big(\hat{p}_{\text{on}}(\cdot \mid s, a), p_{\text{on}}(\cdot \mid s, a)\big) \ \leq \ \sup_{(s,a)} \sqrt{\mathbb{E} \|\widehat{T}(X_0) - T(X_0)\|_2^2}. \tag{35}$$

Combining (35) and (32) gives

$$\sup_{(s,a)} \left| \widehat{\Delta}_{W_2}(s, a) - \Delta_{W_2}(s, a) \right| \ \leq \ \alpha_{N_{\text{on}}, \delta} := \sqrt{K_{\text{stab}} \left( C_0 + C_1 \, W_2(\hat{p}_{\text{off}}, p_{\text{on}}) \right) \Gamma_{N_{\text{on}}, \delta}},$$

with probability at least $1 - \delta$.

*B.3. Capping the on-policy gap for $\hat{\pi}$.* By construction of the replacement rule, for any $(s, a)$ encountered during fine-tuning of $\hat{\pi}$ we have either $\widehat{\Delta}_{W_2}(s, a) \leq \kappa$ (kept offline sample) or we replaced it by an *online* sample. In either case, on those $(s, a)$ we ensure

$$\Delta_{W_2}(s, a) \;\leq\; \widehat{\Delta}_{W_2}(s, a) + \alpha_{N_{\mathrm{on}}, \delta} \;\leq\; \kappa + \alpha_{N_{\mathrm{on}}, \delta}.$$

Therefore the *on-policy* gap of $\hat{\pi}$ obeys

$$\Delta_{W_2}^{(\hat{\pi})} \;\leq\; \kappa + \alpha_{N_{\mathrm{on}}, \delta}. \tag{36}$$

**Step C: Learning-gap bounds for $\pi_{\mathrm{bc}}$ and $\hat{\pi}$ (construction of $\beta$).** Recall the three-term decomposition for any policy $\pi$:

$$\eta_{\mathrm{on}}(\pi_{\mathrm{on}}^\star) - \eta_{\mathrm{on}}(\pi) = \underbrace{\big(\eta_{\mathrm{on}}(\pi_{\mathrm{on}}^\star) - \eta_{\mathrm{off}}(\pi_{\mathrm{on}}^\star)\big)}_{\text{model(a)}} + \underbrace{\big(\eta_{\mathrm{off}}(\pi_{\mathrm{on}}^\star) - \eta_{\mathrm{off}}(\pi)\big)}_{\text{learning(b)}} + \underbrace{\big(\eta_{\mathrm{off}}(\pi) - \eta_{\mathrm{on}}(\pi)\big)}_{\text{model(c)}}. \tag{37}$$

We now bound the *learning* term (b) by ERM generalization.

Policies are learned by ERM on a surrogate imitation loss $\mathcal{L}(\pi)$ over a policy class $\Pi$ with Rademacher complexity $\mathfrak{R}_N(\Pi)$. There exists a calibration constant $C_{\mathrm{val}}$ (depends on concentrability/Lipschitzness; fixed for the class) such that, for any $\pi$,

$$\eta_{\mathrm{off}}(\pi_{\mathrm{on}}^\star) - \eta_{\mathrm{off}}(\pi) \;\leq\; C_{\mathrm{val}}\Big(\mathcal{L}(\pi) - \inf_{\pi' \in \Pi} \mathcal{L}(\pi')\Big). \tag{38}$$

Moreover, for datasets of sizes $N_{\mathrm{off}}, N_{\mathrm{mod}}$, with probability at least $1 - \delta$,

$$\mathcal{L}(\hat{\pi}) - \inf_{\pi' \in \Pi} \mathcal{L}(\pi') \;\leq\; C_\Pi\Big(\mathfrak{R}_N(\Pi) + \sqrt{\tfrac{\log(1/\delta)}{N}}\Big), \qquad N \in \{N_{\mathrm{off}}, N_{\mathrm{mod}}\}, \tag{39}$$

for an absolute constant $C_\Pi$.

Let $\pi_{\mathrm{bc}}$ be the behavior cloning policy trained on the *offline* dataset of size $N_{\mathrm{off}}$, and let $\hat{\pi}$ be ERM on the modified dataset (size $N_{\mathrm{mod}}$): for any $(s, a)$ encountered, if the FM-estimated gap satisfies $\widehat{\Delta}_{W_2}(s, a) > \kappa$, replace the *offline* $(s, a, s')$ sample by an *online* $(s, a, s'_{\mathrm{on}})$ sample (collecting $N_{\mathrm{on}}$ such online transitions), leaving all others unchanged. Denote $N_{\mathrm{mod}}$ the resulting (modified) dataset size.

By calibration (38) and the generalization inequality (39), with probability at least $1 - \delta$ we have

$$\eta_{\mathrm{off}}(\pi_{\mathrm{on}}^\star) - \eta_{\mathrm{off}}(\pi_{\mathrm{bc}}) \;\leq\; C_{\mathrm{val}}\, C_\Pi\Big(\mathfrak{R}_{N_{\mathrm{off}}}(\Pi) + \sqrt{\tfrac{\log(1/\delta)}{N_{\mathrm{off}}}}\Big),$$

and similarly

$$\eta_{\mathrm{off}}(\pi_{\mathrm{on}}^\star) - \eta_{\mathrm{off}}(\hat{\pi}) \;\leq\; C_{\mathrm{val}}\, C_\Pi\Big(\mathfrak{R}_{N_{\mathrm{mod}}}(\Pi) + \sqrt{\tfrac{\log(1/\delta)}{N_{\mathrm{mod}}}}\Big).$$

By a union bound (probability $1 - 2\delta$) and adding these two upper bounds we obtain a common envelope

$$\max\Big\{\eta_{\mathrm{off}}(\pi_{\mathrm{on}}^\star) - \eta_{\mathrm{off}}(\pi_{\mathrm{bc}}),\; \eta_{\mathrm{off}}(\pi_{\mathrm{on}}^\star) - \eta_{\mathrm{off}}(\hat{\pi})\Big\} \;\leq\; \beta, \tag{40}$$

with

$$\beta := C_{\mathrm{val}} C_\Pi\left(\mathfrak{R}_{N_{\mathrm{off}}}(\Pi) + \mathfrak{R}_{N_{\mathrm{mod}}}(\Pi) + \sqrt{\tfrac{\log(2/\delta)}{N_{\mathrm{off}}}} + \sqrt{\tfrac{\log(2/\delta)}{N_{\mathrm{mod}}}}\right). \tag{41}$$

**Step D: Assemble bounds and compare.** Incorporate (27) and (40) to (37) with $\pi = \pi_{\mathrm{bc}}$, we have with high probability,

$$\eta_{\mathrm{on}}(\pi_{\mathrm{on}}^\star) - \eta_{\mathrm{on}}(\pi_{\mathrm{bc}}) \leq \big(\eta_{\mathrm{on}}(\pi_{\mathrm{on}}^\star) - \eta_{\mathrm{off}}(\pi_{\mathrm{on}}^\star)\big) + \beta + \frac{2L_r(1 + \gamma)}{(1 - \gamma)(1 - \gamma L_p)}\Delta_{W_2}$$

Incorporate (27) and (40) to (37) with $\pi = \hat{\pi}$, we have with high probability

$$\eta_{\mathrm{on}}(\pi_{\mathrm{on}}^\star) - \eta_{\mathrm{on}}(\widehat{\pi}) \; \leq \; \big(\eta_{\mathrm{on}}(\pi_{\mathrm{on}}^\star) - \eta_{\mathrm{off}}(\pi_{\mathrm{on}}^\star)\big) + \beta + \frac{2L_r(1+\gamma)}{(1-\gamma)(1-\gamma L_p)}(\kappa + \alpha_{N_{\mathrm{on}},\delta})$$

The difference between these two upper bounds is:

$$\frac{2L_r(1+\gamma)}{(1-\gamma)(1-\gamma L_p)}\big(\Delta_{W_2} - \kappa - \alpha_{N_{\mathrm{on}},\delta}\big).$$

Absorbing $K_{\mathrm{stab}}$ into $C_0$ and $C_1$ gets the result. $\qquad\square$

# I  Proof of Proposition 3.3

We provide a rigorous justification for Proposition 3.3. The key technical point is that the augmented optimal transport (OT) problem on triples $(s, a, s')$ with cost

$$c_\eta\big((s, a, x), (s', a', x')\big) = \|x - x'\|_2^2 + \eta\big(\|s - s'\|_2^2 + \|a - a'\|_2^2\big)$$

reduces, as $\eta \to \infty$, to *conditional* quadratic OT on the next-state variable $x = s'$ given each fixed $(s, a)$, provided that the $(s, a)$-marginals match.

## I.1  Formal Proposition

**Proposition I.1** (Shared-latent coupling recovers conditional $W_2$). **Setup.** *Let* $\mathcal{Y} := \mathcal{S} \times \mathcal{A}$ *denote the conditioning space and* $\mathcal{X} := \mathcal{S}$ *denote the next-state space. Let* $\mu$ *be a probability measure on* $\mathcal{Y}$. *Let* $\alpha, \beta$ *be probability measures on* $\mathcal{Y} \times \mathcal{X}$ *such that their* $\mathcal{Y}$*-marginals coincide:*

$$\pi_{sa}\#\alpha = \pi_{sa}\#\beta = \mu, \tag{42}$$

*where* $\pi_{sa}(y, x) = y$. *Assume* $\alpha$ *and* $\beta$ *admit disintegrations with respect to* $\mu$:

$$\alpha(dy, dx) = \mu(dy)\,\alpha_y(dx), \qquad \beta(dy, dx) = \mu(dy)\,\beta_y(dx), \tag{43}$$

*where* $y = (s, a) \in \mathcal{Y}$ *and* $x \in \mathcal{X}$.

**Augmented OT problem.** *For* $\eta > 0$, *define the augmented quadratic cost on* $(\mathcal{Y} \times \mathcal{X})^2$ *by*

$$c_\eta\big((y, x), (y', x')\big) := \|x - x'\|_2^2 + \eta\|y - y'\|_2^2, \tag{44}$$

*where* $\|y - y'\|_2^2 := \|s - s'\|_2^2 + \|a - a'\|_2^2$. *Define the corresponding OT value*

$$W_{2,\eta}^2(\alpha, \beta) := \inf_{\gamma \in \Pi(\alpha, \beta)} \int c_\eta\big((y, x), (y', x')\big)\, d\gamma, \tag{45}$$

*where* $\Pi(\alpha, \beta)$ *is the set of couplings on* $(\mathcal{Y} \times \mathcal{X})^2$ *with marginals* $\alpha$ *and* $\beta$.

**Assumptions (for** $\eta \to \infty$ **reduction).** *Assume* $\alpha$ *and* $\beta$ *have finite second moments, that* $W_2(\alpha_y, \beta_y) < \infty$ *for* $\mu$*-a.e.* $y$, *and that the conditional squared Wasserstein cost is* $\mu$*-integrable:*

$$\int_{\mathcal{Y}} W_2^2(\alpha_y, \beta_y)\, \mu(dy) < \infty. \tag{46}$$

**Claim 1 (conditional decomposition).** *As* $\eta \to \infty$,

$$\lim_{\eta \to \infty} W_{2,\eta}^2(\alpha, \beta) = \int_{\mathcal{Y}} W_2^2(\alpha_y, \beta_y)\, \mu(dy). \tag{47}$$

**Claim 2 (structure of limiting couplings).** *Let* $\gamma_\eta \in \arg\min_{\gamma \in \Pi(\alpha, \beta)} \int c_\eta\, d\gamma$ *be any sequence of optimal couplings. Then there exists a subsequence* $\gamma_{\eta_n} \Rightarrow \gamma_\infty$ *weakly such that:*

*(i)* $\gamma_\infty$ *is supported on the diagonal set* $\{(y, x; y, x') : y = y'\}$;

*(ii)* $\gamma_\infty$ admits a disintegration

$$\gamma_\infty(dy, dx, dx') = \mu(dy)\,\gamma_y(dx, dx'),$$

where $\gamma_y \in \Pi(\alpha_y, \beta_y)$ is $W_2$-optimal for $\mu$-a.e. $y$.

**OT-FM implication (per fixed conditioning $y$).** Fix any $y = (s, a)$ and let $p_0 = \mathcal{N}(0, I)$. Assume the offline flow satisfies

$$\psi_\theta^{\text{off}}(\cdot, 1 \mid y)\#p_0 = \alpha_y.$$

For $\eta > 0$ and $k \in \mathbb{N}$, let $\psi_\phi^{\text{on},k,\eta}(\cdot, 2 \mid y)$ denote the online flow obtained by running OT-FM to convergence *with minibatch size $k$, where each iteration computes an optimal transport plan between two empirical measures of size $k$ under the* augmented cost (44).

*Assume the standard OT-FM regularity conditions hold for this fixed $y$ (e.g., those in Theorem 4.2 of [47]), including:*

*(A1) $\alpha_y$ and $\beta_y$ have bounded support;*

*(A2) $\alpha_y$ admits a density and the quadratic OT map from $\alpha_y$ to $\beta_y$ exists and is continuous;*

*(A3) the minibatch OT plan is solved exactly at each iteration.*

*Then, for $\mu$-a.e. $y$, we have the two-limit convergence*

$$\lim_{\eta \to \infty} \lim_{k \to \infty} \mathbb{E}_{x_0 \sim p_0}\left[\left\|\psi_\theta^{\text{off}}(x_0, 1 \mid y) - \psi_\phi^{\text{on},k,\eta}\big(\psi_\theta^{\text{off}}(x_0, 1 \mid y), 2 \mid y\big)\right\|_2^2\right] = W_2^2(\alpha_y, \beta_y). \quad (48)$$

## I.2 Proof

*Proof.* We prove (47) by matching an upper bound and a lower bound, which also implies that any limiting optimal coupling concentrates on the diagonal set $\{y = y'\}$. Finally, we connect the resulting conditional OT structure to OT-FM trained with the augmented matching cost (44).

**Step 1 (Upper bound via a diagonal coupling).** Since $\alpha$ and $\beta$ share the same $\mathcal{Y}$-marginal $\mu$, we may couple them by matching the conditioning variable $y$ exactly. For $\mu$-a.e. $y$, let $q_y^\star \in \Pi(\alpha_y, \beta_y)$ be an optimal coupling achieving $W_2^2(\alpha_y, \beta_y)$. A measurable selection $y \mapsto q_y^\star$ can be chosen (see, e.g., [63, Chapter 5] or [52, Section 1.4]). Define $\bar\gamma \in \Pi(\alpha, \beta)$ by

$$\bar\gamma(dy, dx, dy', dx') := \mu(dy)\, q_y^\star(dx, dx')\, \delta_y(dy'), \quad (49)$$

where $\delta_y$ is the Dirac measure at $y$. Because $\bar\gamma$ is supported on $\{y = y'\}$, for every $\eta > 0$,

$$\int c_\eta \, d\bar\gamma = \int_{\mathcal{Y}} \int_{\mathcal{X} \times \mathcal{X}} \|x - x'\|_2^2 \, dq_y^\star(x, x')\, \mu(dy) = \int_{\mathcal{Y}} W_2^2(\alpha_y, \beta_y)\, \mu(dy).$$

Therefore,

$$\limsup_{\eta \to \infty} W_{2,\eta}^2(\alpha, \beta) \le \int_{\mathcal{Y}} W_2^2(\alpha_y, \beta_y)\, \mu(dy). \quad (50)$$

**Step 2 ($y$-transport vanishes as $\eta \to \infty$).** Let $\gamma_\eta$ be an optimal coupling attaining $W_{2,\eta}^2(\alpha, \beta)$ and define

$$C_\star := \int_{\mathcal{Y}} W_2^2(\alpha_y, \beta_y)\, \mu(dy) < \infty.$$

By Step 1, $W_{2,\eta}^2(\alpha, \beta) \le C_\star$, hence

$$\int c_\eta \, d\gamma_\eta = W_{2,\eta}^2(\alpha, \beta) \le C_\star.$$

Since $c_\eta((y, x), (y', x')) \ge \eta \|y - y'\|_2^2$, it follows that

$$\int \|y - y'\|_2^2 \, d\gamma_\eta \le \frac{C_\star}{\eta}. \quad (51)$$

**Step 3 (Extract a weak limit supported on the diagonal).** Because $\alpha$ and $\beta$ have finite second moments, $\Pi(\alpha, \beta)$ is tight, hence $\{\gamma_\eta\}_{\eta>0}$ is tight. Thus there exist $\eta_n \to \infty$ and $\gamma_\infty$ such that $\gamma_{\eta_n} \Rightarrow \gamma_\infty$ weakly. Let $f(y, y') := \|y - y'\|_2^2$ and define the bounded continuous truncation $f_M := \min\{f, M\}$. Since $f_M$ is bounded and continuous,

$$\int f_M \, d\gamma_{\eta_n} \to \int f_M \, d\gamma_\infty.$$

Moreover, $0 \le f_M \le f$, so by (51),

$$\int f_M \, d\gamma_\infty = \lim_{n\to\infty} \int f_M \, d\gamma_{\eta_n} \le \lim_{n\to\infty} \int f \, d\gamma_{\eta_n} = 0.$$

Letting $M \to \infty$ and applying monotone convergence yields $\int f \, d\gamma_\infty = 0$, which implies $y = y'$ holds $\gamma_\infty$-a.s. Hence $\gamma_\infty$ is supported on $\{(y, x; y, x') :\ y = y'\}$.

**Step 4 (Lower bound and convergence of values).** Since $\gamma_\infty$ is supported on $\{y = y'\}$ and has marginals $\alpha$ and $\beta$, it disintegrates as

$$\gamma_\infty(dy, dx, dx') = \mu(dy) \, \gamma_y(dx, dx'),$$

where $\gamma_y \in \Pi(\alpha_y, \beta_y)$ for $\mu$-a.e. $y$. Let $g(x, x') := \|x - x'\|_2^2$ and $g_M := \min\{g, M\}$. Because $0 \le g_M \le c_{\eta_n}$,

$$W_{2,\eta_n}^2(\alpha, \beta) = \int c_{\eta_n} \, d\gamma_{\eta_n} \ge \int g_M \, d\gamma_{\eta_n}.$$

By weak convergence, $\int g_M \, d\gamma_{\eta_n} \to \int g_M \, d\gamma_\infty$. Letting $M \to \infty$ and using monotone convergence yields

$$\liminf_{n\to\infty} W_{2,\eta_n}^2(\alpha, \beta) \ge \int g \, d\gamma_\infty = \int_{\mathcal{Y}} \left( \int \|x - x'\|_2^2 \, d\gamma_y(x, x') \right) \mu(dy) \ge \int_{\mathcal{Y}} W_2^2(\alpha_y, \beta_y) \, \mu(dy).$$

Combining this with (50) proves (47). Moreover, the last inequality can only be tight if

$$\int \|x - x'\|_2^2 \, d\gamma_y(x, x') = W_2^2(\alpha_y, \beta_y) \quad \text{for } \mu\text{-a.e. } y,$$

which implies $\gamma_y$ is a $W_2$-optimal coupling between $\alpha_y$ and $\beta_y$ for $\mu$-a.e. $y$. This proves Claim 2.

**Step 5 (OT-FM with augmented matching cost; then $\eta \to \infty$).** Fix any conditioning value $y = (s, a)$ and recall that the offline flow satisfies

$$\psi_\theta^{\mathrm{off}}(\cdot, 1 \mid y) \# p_0 = \alpha_y.$$

Let $x_1 := \psi_\theta^{\mathrm{off}}(x_0, 1 \mid y)$ with $x_0 \sim p_0$, so that $x_1 \sim \alpha_y$.

*(a) OT-FM population limit for a fixed $\eta$.* Fix $\eta > 0$. Our OT-FM procedure builds minibatch OT matchings between *joint* samples $(y_i, x_i) \sim \alpha$ and $(y'_j, x'_j) \sim \beta$ using the augmented cost $c_\eta$, and then trains a *conditional* map $x \mapsto \psi_\phi^{\mathrm{on},k,\eta}(x, 2 \mid y)$ by regressing onto the matched $x'$ values.

Let $\gamma_\eta \in \Pi(\alpha, \beta)$ be an optimal coupling attaining $W_{2,\eta}^2(\alpha, \beta)$. Disintegrate $\gamma_\eta$ with respect to its first marginal $\alpha$:

$$\gamma_\eta(dy, dx, dy', dx') = \alpha(dy, dx) \, \Gamma_{y,x}^\eta(dy', dx'),$$

where $\Gamma_{y,x}^\eta$ is a probability kernel on $\mathcal{Y} \times \mathcal{X}$. Define the induced conditional law of $x'$ given $(y, x)$ by marginalizing out $y'$:

$$\Lambda_{y,x}^\eta(dx') := \int_{\mathcal{Y}} \Gamma_{y,x}^\eta(dy', dx').$$

The associated barycentric projection is

$$b_\eta(y, x) := \int_{\mathcal{X}} x' \, \Lambda_{y,x}^\eta(dx').$$

Under assumptions (A1)–(A3) and OT-FM convergence guarantees (e.g., Theorem 4.2 of [47], applied with the quadratic cost in the augmented space), training OT-FM to convergence with minibatch size $k$ yields the population limit

$$\lim_{k\to\infty} \mathbb{E}_{(y,x)\sim\alpha}\left[\left\|\psi_\phi^{\mathrm{on},k,\eta}(x,2\mid y) - b_\eta(y,x)\right\|_2^2\right] = 0. \tag{52}$$

In particular, by conditioning on $y$, for $\mu$-a.e. fixed $y$,

$$\lim_{k\to\infty} \mathbb{E}_{x\sim\alpha_y}\left[\left\|\psi_\phi^{\mathrm{on},k,\eta}(x,2\mid y) - b_\eta(y,x)\right\|_2^2\right] = 0. \tag{53}$$

*(b) Send $\eta \to \infty$ (reduction to conditional OT).* Take a subsequence $\eta_n \to \infty$ such that $\gamma_{\eta_n} \Rightarrow \gamma_\infty$ as in Step 3. Let $b_{\eta_n}$ be the barycentric projection induced by $\gamma_{\eta_n}$ as above. Since $\beta$ has finite second moment, $\{b_{\eta_n}\}_n$ is uniformly bounded in $L^2(\alpha)$, so by extracting a further subsequence (not relabeled) we may assume

$$b_{\eta_n} \rightharpoonup b_\infty \quad \text{weakly in } L^2(\alpha). \tag{54}$$

Because $\gamma_\infty$ is supported on $\{y = y'\}$ and disintegrates as

$$\gamma_\infty(dy,dx,dx') = \mu(dy)\,\gamma_y(dx,dx'), \qquad \gamma_y \in \Pi(\alpha_y,\beta_y)\ W_2\text{-optimal},$$

the conditional law of $x'$ given $(y,x)$ under $\gamma_\infty$ is supported on the conditional coupling $\gamma_y$.

Under assumption (A2), for $\mu$-a.e. $y$, the optimal quadratic-cost coupling $\gamma_y$ is induced by a Monge map $T_y : \mathcal{X} \to \mathcal{X}$ (i.e., $x' = T_y(x)$ $\gamma_y$-a.s.). Hence the limiting barycentric projection satisfies

$$b_\infty(y,x) = T_y(x) \quad \text{for } \alpha\text{-a.e. } (y,x).$$

Therefore, for $\mu$-a.e. $y$,

$$\mathbb{E}_{x\sim\alpha_y}\left[\|x - b_\infty(y,x)\|_2^2\right] = \mathbb{E}_{x\sim\alpha_y}\left[\|x - T_y(x)\|_2^2\right] = W_2^2(\alpha_y,\beta_y). \tag{55}$$

Finally, combining (53) with the identification $b_\infty(y,x) = T_y(x)$ and (55), we obtain for $\mu$-a.e. $y$,

$$\lim_{\eta\to\infty}\lim_{k\to\infty} \mathbb{E}_{x\sim\alpha_y}\left[\left\|x - \psi_\phi^{\mathrm{on},k,\eta}(x,2\mid y)\right\|_2^2\right] = W_2^2(\alpha_y,\beta_y).$$

Substituting $x = \psi_\theta^{\mathrm{off}}(x_0,1\mid y)$ with $x_0 \sim p_0$ yields (48). This completes the proof. $\qquad\square$

> **Intuition: Why do we need matched $(s,a)$ marginals?**
>
> When $(s,a)$ is continuous, it is unlikely to observe many online transitions $s'_{\mathrm{on}}$ with exactly the same conditioning pair $(s,a)$. As a result, the OT-FM objective must couple two sets of transitions whose conditioning variables differ across samples. To recover the conditional Wasserstein distance $W_2(p_{\mathrm{off}}(\cdot\mid s,a),\ p_{\mathrm{on}}(\cdot\mid s,a))$ for a fixed $(s,a)$, we add $(s,a)$ into the OT cost:
> $$\|s'_{\mathrm{off}} - s'_{\mathrm{on}}\|_2^2 + \eta\big(\|s_{\mathrm{off}} - s_{\mathrm{on}}\|_2^2 + \|a_{\mathrm{off}} - a_{\mathrm{on}}\|_2^2\big).$$
> As $\eta$ grows, transporting mass across different $(s,a)$ becomes prohibitively expensive, so the optimal coupling is forced to match samples with similar $(s,a)$ and transport occurs primarily in the next-state space.
> This reduction requires the two empirical measures to share the same marginal distribution over $(s,a)$; otherwise, the coupling is forced to move mass in $(s,a)$ even when $\eta$ is large. In practice, we ensure this by sampling $(s_{\mathrm{off}}, a_{\mathrm{off}})$ from the **online** dataset $\mathcal{D}_{\mathrm{on}}$ to construct the offline-generated batch (then sampling $s'_{\mathrm{off}} \sim p_{\mathrm{off}}(\cdot\mid s,a)$ using the offline flow), while sampling online transitions $(s_{\mathrm{on}}, a_{\mathrm{on}}, s'_{\mathrm{on}})$ independently from $\mathcal{D}_{\mathrm{on}}$.

## J  Additional Experimental Results

To further increase the task difficulty, we manually modified the benchmark configurations to amplify the dynamic differences beyond the given settings. For example, in the morphology task, we drastically reduced thigh segment size—making the legs much shorter, in the kinematic task, we almost immobilized the back-thigh joint by severely limiting its rotation range. As shown in Table 2, our method clearly outperforms all other baselines in 5 out of 6 settings, while achieving ties in the remaining one.

| Dataset | Task Name | SAC | BC-SAC | H2O | BC-VGDF | BC-PAR | Ours |
|---------|-----------|-----|--------|-----|---------|--------|------|
| MR | HalfCheetah (Morphology) | $1239 \pm 107$ | $1896 \pm 357$ | $1294 \pm 153$ | $1342 \pm 252$ | $1967 \pm 123$ | $2267 \pm 280$ |
| MR | HalfCheetah (Kinematic) | $2511 \pm 476$ | $4275 \pm 129$ | $3787 \pm 205$ | $3821 \pm 187$ | $3751 \pm 113$ | $4448 \pm 227$ |
| M | HalfCheetah (Morphology) | $1078 \pm 165$ | $1199 \pm 184$ | $1277 \pm 154$ | $1014 \pm 141$ | $1263 \pm 163$ | $1654 \pm 136$ |
| M | HalfCheetah (Kinematic) | $2331 \pm 392$ | $4650 \pm 169$ | $4572 \pm 84$ | $4207 \pm 108$ | $4304 \pm 231$ | $4651 \pm 89$ |
| ME | HalfCheetah (Morphology) | $1115 \pm 207$ | $1067 \pm 208$ | $1034 \pm 206$ | $1048 \pm 202$ | $1196 \pm 203$ | $1834 \pm 193$ |
| ME | HalfCheetah (Kinematic) | $1843 \pm 1185$ | $2650 \pm 447$ | $3739 \pm 196$ | $3409 \pm 207$ | $3108 \pm 500$ | $4537 \pm 74$ |

Table 2: Comparison of return under the extreme difficult settings. MR = Medium Replay, M = Medium, ME = Medium Expert. A cell is green if the method has the highest mean and improves over the second best by at least 2%. Cells within 2% of the top mean are marked in yellow.

# K   Experimental Details of Gym-MuJoCo

In this section, we describe the detailed experimental setup as well as the hyperparameter setup used in this work.

## K.1   Environment Setting

### K.1.1   Offline Dataset

We use the MuJoCo datasets from D4RL [16] as our offline data. These datasets are collected from continuous control environments in Gym [4], simulated using the MuJoCo physics engine [62]. We focus on three benchmark tasks: *HalfCheetah*, *Hopper*, and *Walker2d*, and evaluate across three dataset types: *medium*, *medium-replay*, and *medium-expert*.

- The *medium* datasets consist of trajectories generated by an SAC policy trained for 1M steps and then early stopped.
- The *medium-replay* datasets capture the replay buffer of a policy trained to the performance level of the medium agent.
- The *medium-expert* datasets are formed by mixing equal proportions of medium and expert data (50-50).

### K.1.2   Kinematic Shift Tasks

We use Kinematic Shift Tasks from the benchmark [38]. We select most shift level 'hard' to make the tasks more challenging

• **HalfCheetah Kinematic Shift:** The rotation range of the foot joint is modified to be:

```
<joint axis="0 1 0" damping="3" name="bfoot" pos="0 0 0" range="-.08 .157" stiffness
    ="120" type="hinge"/>
<joint axis="0 1 0" damping="1.5" name="ffoot" pos="0 0 0" range="-.1 .1" stiffness=
    "60" type="hinge"/>
```

• **Hopper Kinematic Shift:** the rotation range of the foot joint is modified from $[-45, 45]$ to $[-9, 9]$:

```
<joint axis="0 -1 0" name="foot_joint" pos="0 0 0.1" range="-9 9" type="hinge"/>
```

• **Walker2D Kinematic Shift:** the rotation range of the foot joint is modified from $[-45, 45]$ to $[-9, 9]$:

```
<joint axis="0 -1 0" name="foot_joint" pos="0 0 0.1" range="-9 9" type="hinge"/>
<joint axis="0 -1 0" name="foot_left_joint" pos="0 0 0.1" range="-9 9" type="hinge"
    />
```

### K.1.3   Morphology Shift Tasks

We use Morphology Shift Tasks from the benchmark [38]. We select most shift level 'hard' to make the tasks more challenging

• **HalfCheetah Morphology Shift:** the front thigh size and the back thigh size are modified to be:

```
<geom fromto="0 0 0.02 0 -0.02" name="bthigh" size="0.046" type="capsule"/>
<body name="fshin" pos="0.02 0 -0.02">
  <geom fromto="0 0 0 -.13 0 -.15" name="bshin" rgba="0.9 0.6 0.6 1" size="0.046"
    type="capsule"/>
</body>
<body name="bfoot" pos="-.13 0 -.15">
  <geom fromto="0 0 0 -.04 0 -0.05" name="fthigh" size="0.046" type="capsule"/>
</body>
<body name="fshin" pos="0 -.04 0 -0.05">
  <geom fromto="0 0 0 .11 0 -.13" name="fshin" rgba="0.9 0.6 0.6 1" size="0.046"
    type="capsule"/>
</body>
<body name="ffoot" pos=".11 0 -.13"/>
```

• **Hopper Morphology Shift:** the foot size is revised to be 0.4 times of that within the source domain:

```
<geom friction="2.0" fromto="-0.052 0 0.1 0.104 0 0.1" name="foot_geom" size="0.024"
    type="capsule"/>
```

• **Walker2D Morphology Shift:** the leg size of the robot is revised to be 0.2 times of that in the source domain.

```
<geom friction="0.9" fromto="0 0 1.05 0 0 0.2" name="thigh_geom" size="0.05" type="
    capsule"/>
<joint axis="0 -1 0" name="leg_joint" pos="0 0 0.2" range="-150 0" type="hinge"/>
<geom friction="0.9" fromto="0 0 0.2 0 0 0.1" name="leg_geom" size="0.04" type="
    capsule"/>
<geom friction="0.9" fromto="0 0 1.05 0 0 0.2" name="thigh_left_geom" rgba=".7 .3 .6
    1" size="0.05" type="capsule"/>
<joint axis="0 -1 0" name="leg_left_joint" pos="0 0 0.2" range="-150 0" type="hinge"
    />
<geom friction="0.9" fromto="0 0 0.2 0 0 0.1" name="leg_left_geom" rgba=".7 .3 .6 1"
    size="0.04" type="capsule"/>
```

### K.1.4 Friction Shift Tasks

Following [38], the friction shift is implemented by altering the friction attribute in the geom elements. The frictional components are adjusted to 5.0 times the frictional components in the offline environment. The following is an example for the Hopper robot.

Listing 1: Geometry Definitions for Walker2D

```
# torso
<geom friction="4.5" fromto="0 0 1.45 0 0 1.05" name="torso_geom" size="0.05" type="
    capsule"/>
# thigh
<geom friction="4.5" fromto="0 0 1.05 0 0 0.6" name="thigh_geom" size="0.05" type="
    capsule"/>
# leg
<geom friction="4.5" fromto="0 0 0.6 0 0 0.1" name="leg_geom" size="0.04" type="
    capsule"/>
# foot
<geom friction="10.0" fromto="-0.13 0 0.1 0.26 0 0.1" name="foot_geom" size="0.06"
    type="capsule"/>
```

## K.2 Implementation Details

**BC-SAC:** This baseline leverages both offline and online transitions for policy learning. Since learning from offline data requires conservatism while online data does not, we incorporate a behavior cloning term into the actor update of the SAC algorithm. Specifically, the critic is updated using

standard Bellman loss on the combined offline and online datasets, and the actor is optimized as:

$$\mathcal{L}_{\text{actor}} = \lambda \cdot \mathbb{E}_{s \sim \mathcal{D}_{\text{off}} \cup \mathcal{D}_{\text{on}}, \ a \sim \pi_\varphi(\cdot|s)} \left[ \min_{i=1,2} Q_{\varsigma_i}(s,a) - \alpha \log \pi_\varphi(\cdot|s) \right] + \mathbb{E}_{(s,a) \sim \mathcal{D}_{\text{off}}, \ \hat{a} \sim \pi_\varphi(\cdot|s)} \left[ (a - \hat{a})^2 \right],$$

(56)

where $\lambda = \frac{\omega}{\frac{1}{N} \sum_{(s_j,a_j)} \min_{i=1,2} Q_{\varsigma_i}(s_j,a_j)}$ and $\omega \in \mathbb{R}^+$ is a normalization coefficient. We train BC-SAC for 400K gradient steps, collecting online data every 10 steps. We use the hyperparameters recommended in [38].

**H2O:** H2O [46] trains domain classifiers to estimate dynamics gaps and uses them as importance sampling weights during critic training. It also incorporates a CQL loss to encourage conservatism. Since the original H2O is designed for the Online-Offline setting, we adapt the objective to the Offline-Online setting. The critic loss is:

$$\mathcal{L}_{\text{critic}} = \mathbb{E}_{\mathcal{D}_{\text{on}}} \left[ (Q_{\varsigma_i}(s,a) - y)^2 \right] + \mathbb{E}_{\mathcal{D}_{\text{off}}} \left[ \omega(s,a,s')(Q_{\varsigma_i}(s,a) - y)^2 \right]$$
$$+ \beta_{\text{CQL}} \left( \mathbb{E}_{s \sim \mathcal{D}_{\text{off}}, \ \hat{a} \sim \pi_\varphi(\cdot|s)} \left[ \omega(s,a,s') Q_{\varsigma_i}(s,\hat{a}) \right] - \mathbb{E}_{\mathcal{D}_{\text{off}}} \left[ \omega(s,a,s') Q_{\varsigma_i}(s,a) \right] \right), \quad i \in \{1,2\},$$

(57)

where $\omega(s,a,s')$ is the dynamics-based importance weight, and $\beta_{\text{CQL}}$ is the penalty coefficient. We set $\beta_{\text{CQL}} = 10.0$, which performs better than the default 0.01. We reproduce H2O using the official codebase,[3] and adopt the suggested hyperparameters. H2O is trained for 40K environment steps, with 10 gradient updates per step.

**BC-VGDF:** BC-VGDF [65] filters offline transitions whose estimated values align closely with those from the online environment. It trains an ensemble of dynamics models to predict next states from raw state-action pairs under the online dynamics. Each predicted next state is evaluated by the policy to obtain a value ensemble $\{Q(s_i', a_i')\}_{i=1}^{M}$, forming a Gaussian distribution. A fixed percentage ($\xi\%$) of offline transitions with the highest likelihood under this distribution are retained. The critic loss is:

$$\mathcal{L}_{\text{critic}} = \mathbb{E}_{(s,a,r,s') \sim \mathcal{D}_{\text{on}}} \left[ (Q_{\varsigma_i}(s,a) - y)^2 \right]$$
$$+ \mathbb{E}_{(s,a,r,s') \sim \mathcal{D}_{\text{off}}} \left[ \mathbf{1} \left( \Lambda(s,a,s') > \Lambda_{\xi\%} \right) (Q_{\varsigma_i}(s,a) - y)^2 \right], \quad i \in \{1,2\}, \quad (58)$$

where $\Lambda(s,a,s')$ denotes the fictitious value proximity (FVP), and $\Lambda_{\xi\%}$ is the $\xi$-quantile threshold. VGDF also trains an exploration policy. Actor training includes a behavior cloning term, as in BC-SAC. We follow the official implementation,[4] use the recommended hyperparameters, and train for 40K environment steps with 10 gradient updates per step.

**BC-PAR:** BC-PAR [37] addresses dynamics mismatch through representation mismatch, measured as the deviation between the encoded source state-action pair and its next state. It employs a state encoder $f_\psi$ and a state-action encoder $g_\xi$, both trained on the target domain. The encoder loss is:

$$\mathcal{L}(\psi, \xi) = \mathbb{E}_{(s,a,s') \sim \mathcal{D}_{\text{on}}} \left[ (g_\xi(f_\psi(s), a) - \text{SG}(f_\psi(s')))^2 \right], \quad (59)$$

where SG is the stop-gradient operator. Rewards of the offline data are adjusted as:

$$\hat{r}_{\text{PAR}} = r_{\text{off}} - \beta \cdot \|g_\xi(f_\psi(s_{\text{off}}), a_{\text{off}}) - f_\psi(s_{\text{off}}')\|^2, \quad (60)$$

where $\beta$ controls the penalty strength. The actor ($\pi_\varphi$) and critic ($Q_{\varsigma_i}$) are jointly trained using both offline and online data. Actor training includes a behavior cloning term, similar to BC-SAC. We implement BC-PAR using the official codebase,[5] adopt the suggested hyperparameters, and train for 40K environment steps with 10 gradient updates per step.

COMPFLOW. When training the target flow, we use a quadratic cost function and employ the Python Optimal Transport (POT) library [15] to compute the optimal transport plan for each minibatch using the `exact` solver. Additional hyperparameters are provided in Table 3 and Table 4. Since the exploration bonus term is closely tied to properties of the environment—such as the state space, action space, and reward structure—it is expected that the optimal exploration strength $\beta$ varies across tasks. We perform a sweep over $\beta \in \{0.01, 0.1, 0.2\}$, and select the offline data selection ratio $\xi\%$ from $\{30\%, 50\%\}$.

---

[3] https://github.com/t6-thu/H2O
[4] https://github.com/Kavka1/VGDF
[5] https://github.com/dmksjfl/PAR

| Hyperparameter | Value |
|---|---|
| Actor network architecture | (256, 256) |
| Critic network architecture | (256, 256) |
| Batch size | 128 |
| Learning rate | $3 \times 10^{-4}$ |
| Optimizer | Adam [25] |
| Discount factor ($\gamma$) | 0.99 |
| Replay buffer size | $10^6$ |
| Warmup steps | $10^5$ |
| Activation | ReLU |
| Target update rate | $5 \times 10^{-3}$ |
| SAC temperature coefficient ($\alpha$) | 0.2 |
| Maximum log standard deviation | 2 |
| Minimum log standard deviation | $-20$ |
| Normalization coefficient ($\omega$) | 5 |

Table 3: Hyperparameters for RL training.

| Hyperparameter | Offline Flow | Online Flow |
|---|---|---|
| Number of hidden layers | 6 | 6 |
| Hidden dimension | 256 | 256 |
| Activation | ReLU | ReLU |
| Batch size | 1024 | 1024 |
| ODE solver method | Euler | Euler |
| ODE solver steps | 10 | 10 |
| Training frequency | — | 5000 |
| Optimizer | Adam | Adam |

Table 4: Hyperparameter setup for the offline and online flows

## L   Experimental Details of Wildlife Conservation

We use the green security simulator in [67]. The model is a Markov decision process with state, action, transitions, and a terminal return. We summarize the parts needed to reproduce our experiments.

**State and action.** At time $t$, the state is

$$s_t = (a_{t-1}, w_{t-1}, t), \qquad s_0 = (0, w_0, 0),$$

where $w_t = (w_t^1, \ldots, w_t^N)$ is wildlife across $N$ cells and $a_t = (a_t^1, \ldots, a_t^N)$ is patrol effort. The defender chooses $a_t \in [0, 1]^N$ under the budget $\sum_{i=1}^N a_t^i \leq B$.

**Adversary behavior.** At each step, the poacher places a snare in cell $i$ with probability

$$p_t^i = \text{logistic}\left(z^i + \beta a_{t-1}^i + \eta \sum_{j \in \mathcal{N}(i)} a_{t-1}^j\right),$$

where $z^i$ is the baseline attractiveness of cell $i$. The parameters $\beta < 0$ and $\eta > 0$ capture deterrence from prior patrol and displacement from neighboring patrols. The realized attack is

$$k_t^i \sim \text{Bernoulli}(p_t^i).$$

**Wildlife transition.** After attacks, wildlife in each cell evolves by natural growth and poaching losses:

$$w_t^i = \max\left\{0, \left(w_{t-1}^i\right)^\phi - \alpha k_{t-1}^i \left(1 - a_t^i\right)\right\},$$

where $\phi > 1$ is the growth rate and $\alpha > 0$ is the loss per uncovered attack. This defines the transition kernel $T_z$ over states given actions:

$$s_{t+1} \sim T_z(s_t, a_t).$$

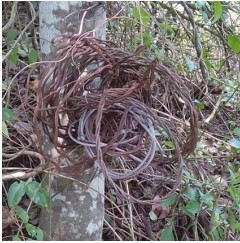 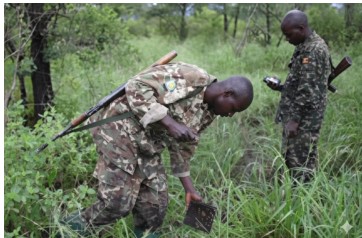

Figure 7: Well-hidden snares and rangers conducting a patrol to locate them. Photos credit: Uganda Wildlife Authority

**Return.** The episode return is the total wildlife at the horizon,

$$R(s_T) \;=\; \sum_{i=1}^{N} w_T^i,$$

and the expected return of a policy $\pi$ under environment parameters $z$ is

$$r(\pi, z) \;=\; \mathbb{E}[R(s_T)], \qquad s_{t+1} \sim T_z\big(s_t, \pi(s_t)\big), \; s_0 = (0, w_0, 0).$$

We assume access to an offline dataset of 100,000 transitions collected in Murchison Falls National Park using a well-trained SAC policy. For the online environment in Queen Elizabeth National Park, we are allowed a budget of 40,000 interactions.

## M Discussion on Computational Cost

The practical cost of data filtering is relatively small. For example, with a batch size of 256 for training the policy network and a Monte Carlo sample size of 30, the entire filtering process takes just 0.03 seconds on an A100 GPU. We will highlight in the paper that this efficiency is due to the simplicity of our flow training objective, which follows a linear path. As a result, solving the corresponding ODE at inference time is very easy—A basic Euler method with just 10 time steps is sufficient.

