# OpenReview forum: "Composite Flow Matching for Reinforcement Learning with Shifted-Dynamics Data"
_NeurIPS.cc/2025/Conference — NeurIPS 2025 spotlight_

### Official Review · Reviewer_DanK · 2025-06-26

**Clarity:** 2
**Significance:** 3
**Originality:** 2
**Rating:** 4
**Confidence:** 5

**Summary:**

This paper proposes a method that leverages the source offline datasets to improve sample efficiency and generalization in the target environment. Specifically, the method uses optimal transport to calculate the dynamic gap between the source task and the target task, and then selectively expands the replay buffer of the target task for agent training.

**Questions:**

1. I suggest that the author specify the range of values for t in the source flow and target flow in Section 3.1.
2. To address the issue of data scarcity (mentioned in line 136) in the target domain, the author proposes using a two-stage flow to model the target data distribution (line 148), namely, first flowing from the Gaussian prior to the source data distribution, and then from the target data distribution. However, the loss function from the source data distribution to the target data distribution is still based on a small amount of target data, and I believe the issue of data scarcity still persists.
3. Why is the loss function of the target flow shown in line 174 inconsistent with the loss function in line 16, Algorithm 3?
4. In Proposition 3.2, suppose both the source flow and target flow can perfectly model the flow map, according to the content introduced by the author in Section 2.3, the corresponding Wasserstein cost for this formula is $(s'_{tar} - s'_{src})^2$. However, the author has mentioned that the Wasserstein cost they intend to use is the cost shown in line 172. Therefore, the author needs to clarify this point.
5. Essentially, Equation (2) measures how large the gap is between s' that the source dynamics and target dynamics transition to under the same conditions (s, a). Is this understanding correct?
6. Theorem 3.3 is counterintuitive because, intuitively, selecting samples from the source data that are completely dissimilar to the target data will clearly hinder the model's generalization rather than improve it. Therefore, I am skeptical about the conclusion of Theorem 3.3.
7. Since Theorem 3.3 mentions that 'exploration in regions with high dynamics gap can reduce the performance gap relative to the optimal policy,' why, in practice, do the authors choose the source data with the lowest gap?
8. To fully demonstrate the generalization performance of the proposed method, I suggest that the author conduct tests on source and target tasks with larger dynamic differences, such as training on HalfCheetah and testing the generalization performance on Walker2D. Alternatively, the author could consider environments in Meta-World [1], which have the same state and action spaces.

    [1] Meta-world: A benchmark and evaluation for multi-task and meta reinforcement learning.
9. I suggest that the author compare the performance differences between direct flow and composite flow in ablation study, rather than comparing the MSE of the loss, as the loss may be influenced by the values of each dimension in the state and action space.
10. How is $v^*$ obtained in line 12 of Algorithm 1? Additionally, could the author explain the meaning of $A(i,j)$ and provide a detailed calculation of the optimal transport plan's solution process?
11. Does the author intend to express that x represents a transition in line 10 of Algorithm 2? However, the v in the loss function provided here is an unconditional velocity field, while the source flow loss in Section 3.1 actually refers to a conditional velocity field.

**Ethical Concerns:**

["NO or VERY MINOR ethics concerns only"]

**Final Justification:**

The authors have addressed all of my concerns

**Limitations:**

yes

**Paper Formatting Concerns:**

The caption of the table should be placed above the table.

**Quality:**

2

**Strengths And Weaknesses:**

Advantages: The idea presented in this paper is innovative and the theory is solid.

Disadvantages: Some parts of the paper are vaguely expressed, which could cause confusion for readers. Please refer to the issues in the question section and improve the article accordingly.

---

> ### Author Rebuttal · Authors · 2025-07-31
>
> We thank the reviewer for acknowledging the novelty of our idea and the soundness of our theoretical contributions. However, we believe several confusions are caused by a misunderstanding regarding our framework, specifically Theorem 3.3: we do **not** add dissimilar source data to the replay buffer. Instead, we keep offline source data in low-gap regions and explore high-gap regions in the **target** domain, and Theorem 3.3 is consistent with implementation. We will clarify this in our revised manuscript. Below, we provide detailed responses to the questions.
>
> **Q1. I suggest that ... in Section 3.1.**
>
> The time $t$ for the source flow is [0,1], and for the target flow is [1,2], as specified in lines 145 and 149, respectively. We will clarify this more clearly in the revised manuscript
>
> **Q2. To address the issue of data scarcity ... data scarcity still persists.**
>
> Our goal is to develop an algorithm that achieves better performance using only a small amount of data, compared to directly training a flow matching model from scratch. While the performance of composite flow also degrades when $N_{\rm tar}$​ is small, our focus is on demonstrating **relative improvements** over the direct flow baseline, rather than claiming to fully solve the problem.
>
> Both theoretically and empirically, we demonstrate that initializing from the source domain distribution provides a more informative prior than training the flow model from scratch. This initialization leverages prior knowledge from the source domain, resulting in improved performance when target data is limited.
>
> **Q3. Why is the loss function ... in line 16, Algorithm 3?**
>
> Thank you for pointing out the typo in line 174. The right-hand side should be written as $(s_{\rm tar}' - s_{\rm src}')$.
>
> Regarding Algorithm 3, line 16: the variable $v^*$ is defined just before line 15, and the variables $x_1$ and $x_2$ correspond to $s_{\rm src}'$ and $ s_{\rm tar}'$, as defined in lines 5 and 8, respectively. Therefore, the two expressions are equivalent after fixing the typo.
>
> We will correct this typo in our revised manuscript.
>
> **Q4. In Proposition 3.2, suppose ... Therefore, the author needs to clarify this point.**
>
> The cost function in Section 2.3 is for flow matching without conditional variables (s, a in our case). As explained in lines 165–172, we are performing conditional variable flow matching for modeling $p(s'|s,a)$. Therefore, we use the cost function in line 172, which follows the formulation proposed in [17]. We will clarify this distinction more clearly in the revised manuscript.
>
> **Q5. Essentially, Equation (2) ... Is this understanding correct?**
>
> Yes.
>
> **Q6. Theorem 3.3 is counterintuitive because, intuitively, selecting samples from the source data that are completely dissimilar to the target data will clearly hinder the model's generalization rather than improve it. Therefore, I am skeptical about the conclusion of Theorem 3.3.**
>
> We would like to clarify that we do **not** add dissimilar source domain data to the replay buffer. As defined in Section 2.1, our setting assumes a **fixed offline source** dataset, and we do not perform any exploration in the source domain.
>
> Theorem 3.3 states that if we **retain** only **low**-dynamics-gap source data in the replay buffer and actively explore high-dynamics regions of the **target** domain to replace dissimilar source data, overall performance will improve. In this process, dissimilar source domain data is filtered out.
>
> The benefit of exploring high-gap regions in the target domain is intuitive: since the replay buffer already includes source samples from low-gap regions, exploring high-gap areas in the target domain complements the dataset and enhances overall coverage.
>
> We will clarify this more explicitly in our revised manuscript.
>
>
> **Q7. Since Theorem 3.3 mentions that 'exploration in regions with high dynamics gap can reduce the performance gap relative to the optimal policy,' why, in practice, do the authors choose the source data with the lowest gap?**
>
> We believe this confusion is potentially caused by a misunderstanding of the statement of Theorem 3.3, which is clarified in Q6.
>
> To summarize, the exploration of the high dynamics gap regions in Theorem 3.3 takes place in the **target** domain. The source domain consists of a fixed offline dataset, and **no** exploration is performed in the source domain. Our approach selectively uses offline source data in low-gap regions while actively exploring high-gap regions in the target domain. Therefore, the setup in Theorem 3.3 is fully consistent with our implementation.
>
> We will make this clarification more explicit in our revised manuscript.
>
> **Q8. To fully demonstrate ... which have the same state and action spaces.**
>
> Thank you for the suggestion. The tasks used in our paper are drawn from a public benchmark in this area [31], where each task includes three levels of difficulty. In our experiments, we already adopt the most challenging level, which corresponds to the largest dynamics gap within the benchmark.
>
> Following your suggestion to further increase the task difficulty, we manually modified the benchmark configurations to amplify the dynamic differences beyond the given settings. For example, in the morphology task, we drastically reduced thigh segment size—making the legs much shorter, in the kinematic task, we almost immobilized the back-thigh joint by severely limiting its rotation range. Our algorithm clearly outperforms all other baselines in 4 out of 5 settings, while achieving a tie in the remaining one.
>
> | Dataset | Task Name                | SAC           | BC_SAC       | H2O          | BC_VGDF      | BC_PAR       | Ours         |
> |---------|--------------------------|---------------|--------------|--------------|--------------|--------------|--------------|
> | MR      | HalfCheetah (Morphology) | 1239 (107)    | 1896 (357)   | 1294 (153)   | 1342 (252)   | 1967 (123)   | **2267 (280)**   |
> | MR      | HalfCheetah (Kinematic)  | 2511 (476)    | 4275 (129)   | 3787 (205)   | 3821 (187)   | 3751 (113)   | **4448 (227)**   |
> | M       | HalfCheetah (Morphology) | 1078 (165)    | 1199 (184)   | 1277 (154)   | 1014 (141)   | 1263 (163)   | **1654 (136)**   |
> | M       | HalfCheetah (Kinematic)  | 2331 (392)    | 4650 (169)   | 4572 (84)    | 4207 (108)   | 4304 (231)   | 4651 (89)    |
> | ME      | HalfCheetah (Morphology) | 1115 (207)    | 1067 (208)   | 1034 (206)   | 1048 (202)   | 1196 (203)   | **1834 (193)**   |
> | ME      | HalfCheetah (Kinematic)  | 1843 (1185)   | 2650 (447)   | 3739 (196)   | 3409 (207)   | 3108 (500)   | **4537 (74)**    |
>
> *MR = Medium-Replay, M = Medium, ME = Medium-Expert*
>
> Regarding your suggestion on HalfCheetah and Walker2D: while these tasks differ significantly in dynamics, they also differ in the physical meaning of their state and action spaces. Our method is designed for settings where the state and action spaces are consistent and only the transition dynamics vary.
>
> As for Meta-World, although tasks have different goals and objects, they share the same underlying transition dynamics $p(s'|s,a)$, which differs from our setting where the source and target domains have distinct transition models. Our formulation (lines 74–80) specifically targets scenarios with varying dynamics while keeping the rest of the MDP structure fixed.
>
>
> **Q9. I suggest that the author ... in the state and action space.**
>
> We report the reward values for Direct Flow and Composite Flow in the table below, using the medium offline dataset. As shown, Composite Flow consistently achieves lower reward compared to Direct Flow. Note that Direct Flow cannot compute the Wasserstein distance between the source and target transition probabilities using Proposition 1. Instead, we estimate the Wasserstein distance for a given $(s,a)$ pair by sampling 100 transitions from the source and target flow respectively.
>
> | Task             | CompFlow | DirectFlow |
> |------------------|------------------------|---------------------------|
> | HalfCheetah Morph | **2282 ± 287**             | 1538 ± 328                |
> | Hopper Morph      | **604 ± 173**              | 442 ± 65                 |
> | Walker2D Morph    | **886 ± 372**              | 532 ± 314                 |
>
> **Q10. How is $v^*$ obtained ... solution process?**
>
> $v^*$ is computed in line 10 as the difference between each pair of initial and terminal points.
>
> $A$ is the optimal transport plan, represented as a doubly stochastic matrix. Each entry $A(i,j)$ denotes the fraction of mass transported from source sample $i$ to target sample $j$. A larger value of $A(i,j)$ indicates a higher probability that the pair $(i,j)$ will be sampled together.
>
> To compute $A$, we solve the following linear program:
> $$\min_A \sum_{i,j} A(i,j) \cdot c(i,j) \quad \text{subject to} \quad A \mathbf{1} = \mu, \quad A^\top \mathbf{1} = \nu, \quad A(i,j) \geq 0$$
> where $c(i,j)$ is the cost function, and $\mu, \nu$ are uniform distributions over the source and target samples, respectively. We implement this using the POT library. We will clarify this in the revised manuscript.
>
>
> **Q11. Does the author intend ... actually refers to a conditional velocity field.**
>
> Yes, $x_t^{(i)}$ in line 10 is a linear interpolation between the initial point $x_0^{(i)}$ and the final point $x_1^{(i)} \doteq s'^{(i)}$. The velocity field $v$ depends on the sampled $s'$, which itself is drawn from the conditional distribution $p(s'|s, a)$ given a specific $(s, a)$ pair. Therefore, although not explicitly written, the velocity field $v$ is indeed conditioned on $(s, a)$, making it a conditional velocity field.
>
> The source flow training follows the standard framework of conditional variable flow matching as established in prior literatures—see, for example, Eq. (5) in (Park et al., 2025)
>
> *Seohong Park and Qiyang Li and Sergey Levine. ''Flow Q-learning.'' The Forty-second International Conference on Machine Learning, 2025.*

---

> > ### Comment · Reviewer_DanK · 2025-08-05
> >
> > I appreciate the clarification and responses provided by the authors, and I hope they will revise all unclear expressions in the article mentioned in the revised manuscript. I am willing to increase my score as recognition of the authors' hard work on the rebuttal.

---

> ### Author Response · Authors · 2025-08-06
>
> We thank the reviewer for the encouragement and positive feedback. We greatly appreciate the comments, which help strengthen our paper! We will incorporate the clarifications and additional experimental results in the final version.

---

### Official Review · Reviewer_fhWu · 2025-06-29

**Clarity:** 3
**Significance:** 4
**Originality:** 3
**Rating:** 5
**Confidence:** 4

**Summary:**

The paper tackles the offline-to-online RL setting in which the available offline data come from environments with shifted dynamics. It introduces a novel method for quantifying the dynamics gap that combines a composite-flow model with the Wasserstein distance. The approach shows strong empirical performance on a set of simulated tasks and, in my view, offers a timely contribution to the field.

**Questions:**

- **Assumptions on the source data.** What coverage properties are required? How many source samples are needed for a given state–action pair?
- **Pure online RL baseline.** To demonstrate that the source data do not hinder learning in the target environment, a comparison with a policy trained from scratch solely on target-environment data will be appreciated.

**Ethical Concerns:**

["NO or VERY MINOR ethics concerns only"]

**Final Justification:**

The authors addressed the concerns and they showcased publishable material at Neurips.

**Limitations:**

-na-

**Paper Formatting Concerns:**

The paper format looks to adhere to NeurIPS style.

**Quality:**

3

**Strengths And Weaknesses:**

Strengths:
- The idea of using composite flow models and the Wasserstein metric to estimate the dynamics gap is novel.
- The empirical evaluation is thorough; the proposed method consistently outperforms relevant baselines across most tasks.

    Weaknesses:
- Wasserstein distances are approximated by solving discrete OT problems on mini-batches, which adds computational overhead and approximation error. The impact of mini-batch size is not analyzed.
- **Theorem 3.1**
    - The bound on the total-variation distance lacks explicit rate information: does it tighten as $N_{tar}$ increases?
    - The bound can exceed 1, which makes it vacuous; please clarify the conditions under which it is non-trivial.
    - Because the rest of the paper measures gaps with the Wasserstein distance, presenting this bound in the same metric would improve consistency.
- **Theorem 3.3**
    - Is $\Delta_\kappa$ always non-negative? If not, under what conditions is it positive?

---

> ### Author Rebuttal · Authors · 2025-07-31
>
> We thank the reviewer for recognizing the novelty of the proposed composite flow to estimate the dynamics gap and for acknowledging the thoroughness of our evaluation. Below, we provide detailed responses to the questions.
>
> **Q1. Wasserstein distances are approximated by solving discrete OT problems on mini-batches, which adds computational overhead and approximation error. The impact of mini-batch size is not analyzed.**
>
> The OT problems are efficiently solved using the Python POT package, which provides a multi-threaded CPU implementation. For example, with a batch size of 1024, solving an OT problem takes us only 0.18 seconds. Importantly, OT is only required to be solved during the training phase of the target flow. Once training is complete, estimating the dynamics gap as in Eq.2 for a given $(s, a)$ pair using the trained flows no longer requires solving OT problems.
>
> Additionally, we provide below the reward versus OT coupling batch size on the HalfCheetah-Medium replay environment across different shift tasks:
>
> | Batch Size | 8            | 256          | 512          | 1024         | 2048         |
> |------------|--------------|--------------|--------------|--------------|--------------|
> | Morphology | 2508 ± 83    | 2893 ± 152   | 3097 ± 123   | 3119 ± 107   | 3258 ± 119   |
> | Kinematic  | 4607 ± 323   | 4801 ± 265   | 4992 ± 354   | 5189 ± 262   | 5096 ± 287   |
>
> As shown, performance stabilizes once the batch size exceeds 1024. Therefore, we use a batch size of 1024 for the OT coupling in all experiments.
>
>
> **Q2.1. The bound on the total-variation distance lacks explicit rate information: does it tighten as $N_{\rm tar}$ increases?**
>
> Yes, as $N_{\rm tar}$​ increases, the bound becomes tighter, meaning that the requirement on the similarity between the source and target domains becomes stricter. This is intuitive: when abundant target domain data (large $N_{\rm tar}$) is available, the benefit of incorporating source domain data diminishes, as external knowledge is no longer necessary. In our setting, however, we assume limited target domain data (small $N_{\rm tar}$), making the bound easier to satisfy. We will clarify this point more explicitly in the revised manuscript.
>
> **Q2.2. The bound can exceed 1, which makes it vacuous; please clarify the conditions under which it is non-trivial.**
>
>
> We have state variance in the numerator, which is bounded by the square of state norm bound ($\zeta^2$) in the denominator. Therefore, these two terms together derive a value smaller than 1. The remaining terms left are the constant $C$ in the numerator and the square root of dataset sizes in the denominator. $C$ is a constant related to the 4-th moment of the state distribution, which depends on the target domain environment itself instead of the collected dataset. Hence, having both dataset sizes larger than $C^2$ is a sufficient condition to ensure that the bound does not exceed 1. We will include this discussion in our revised manuscript.
>
>
> **Q2.3. Because the rest of the paper measures gaps with the Wasserstein distance, presenting this bound in the same metric would improve consistency.**
>
> Thanks for this nice suggestion. Indeed, Theorem 3.1 can be adapted using the Wasserstein distance. The condition for Wasserstein distance is the same as the condition for TV distance under the big O notation. The only difference lies in the constant factor, which is absorbed by the big-O. We will update the bound to use Wasserstein distance in our revised manuscript.
>
>
> **Q3. Is $\Delta_{\kappa}$ always non-negative? If not, under what conditions is it positive?**
>
> $\Delta_{\kappa}$ is generally non-negative when the estimation error of the dynamic gap or $\kappa$ is small. Although the estimation error of the dynamics gap is inevitable—particularly when $N_{\rm tar}$ is small—we mitigate its impact through a warm-up phase, as shown in Table 3 of the appendix, where data is collected without exploration to reduce initial large estimation error. After this warm-up phase, we can choose an appropriate value of $\kappa$ to ensure that $\Delta_{\kappa}$ remains non-negative.
>
>
> **Q4. Assumptions on the source data. What coverage properties are required? How many source samples are needed for a given state–action pair?**
>
> *What coverage properties are required?*
>
> Offline RL algorithms typically require some concentration conditions for performance guarantee. A common assumption is the existence of a constant $C_\star=\sup_{s,a}\frac{d_{\pi_\star}(s,a)}{\mu(s,a)}<\infty$, where $d_{\pi_\star}$ is the occupancy of the optimal policy $\pi_\star$ and $\mu$ is the empirical state–action distribution of the dataset $D$.
>
> In contrast, our method allows for online exploration in the target domain. Consequently, optimality no longer depends entirely on the coverage of the source data. With improved coverage, the composite flow produces more accurate estimates. Therefore, by Theorem 3.3, we obtain a tighter bound on the performance gap relative to the optimal policy. We will include this analysis in the revised manuscript.
>
> *How many source samples are needed for a given state–action pair?*
>
> Theoretically, when the number of source samples for a given state–action pair increases, the estimation of the dynamics gap will be more accurate. As a result, according to Theorem 3.3, the performance gap from the optimal policy will be reduced, indicating we get a better policy in the target domain.
>
> Empirically, our tasks involve continuous state and action spaces, making it infeasible to obtain multiple samples for an exact state-action pair. To better understand the dataset’s coverage, we compute the number of samples falling within Euclidean balls (of varying radii) centered at the mean of (s, a) pairs.
>
>
> | Dataset                         | Distance < 2 | Distance < 2.5 | Distance < 3 | Distance < 3.5 | Distance < 4 | Distance < 4.5 | Distance < 6 | Distance < 8 |
> |---------------------------------|-------------:|---------------:|------------:|---------------:|------------:|---------------:|------------:|-------------:|
> | HalfCheetah-Medium              |            0 |              2 |          56 |            267 |         800 |           1828 |       10785 |        39325 |
> | HalfCheetah-Medium-Replay       |            0 |              2 |          22 |             99 |         457 |            805 |        3867 |        12455 |
> | HalfCheetah-Medium-Expert       |            0 |              0 |           0 |              8 |          74 |            369 |        5949 |        42164 |
>
>
> **Q5. Pure online RL baseline. To demonstrate that the source data do not hinder learning in the target environment, a comparison with a policy trained from scratch solely on target-environment data will be appreciated.**
>
> We provide the performance of SAC trained from scratch in the target domain below. As expected, its reward is significantly lower than our method.
>
> | Dataset | Task Name                   | SAC              | Ours             | Highest                       |
> |---------|-----------------------------|------------------|------------------|-------------------------------|
> | MR  | HalfCheetah (Morphology)    | 1457 ± 89        | 3119 ± 107       | Ours                          |
> | MR  | HalfCheetah (Kinematic)     | 2255 ± 197       | 5189 ± 262       | Ours                          |
> | MR  | Hopper (Morphology)         | 364 ± 82         | 355 ± 6          | SAC                           |
> | MR  | Hopper (Kinematic)          | 737 ± 547        | 1024 ± 1         | 1025 ± 0 (H2O)                |
> | MR  | Walker2D (Morphology)       | 253 ± 60         | 1094 ± 791       | Ours                          |
> | M   | HalfCheetah (Morphology)    | 1467 ± 89        | 2282 ± 287       | Ours                          |
> | M   | HalfCheetah (Kinematic)     | 2316 ± 92        | 5593 ± 44        | Ours                          |
> | M   | Hopper (Morphology)         | 396 ± 60         | 604 ± 173        | Ours                          |
> | M   | Hopper (Kinematic)          | 724 ± 535        | 1023 ± 2         | Ours                          |
> | M   | Walker2D (Morphology)       | 301 ± 177        | 886 ± 372        | Ours                          |
> | ME  | HalfCheetah (Morphology)    | 1392 ± 238       | 1485 ± 67        | Ours                          |
> | ME  | HalfCheetah (Kinematic)     | 2323 ± 97        | 5750 ± 84        | Ours                          |
> | ME | Hopper (Morphology)         | 359 ± 75         | 462 ± 89         | Ours                          |
> | ME  | Hopper (Kinematic)          | 724 ± 535       | 1022 ± 2         | 1031 ± 3 (H2O)                |
> | ME  | Walker2D (Morphology)       | 228 ± 51         | 648 ± 180        | 1103 ± 444 (H2O)              |
>
> *MR = Medium Replay, M = Medium, ME = Medium Expert.*

---

> > ### Comment · Reviewer_fhWu · 2025-08-06
> >
> > Thanks for the detailed clarification! The authors' response addressed my questions well, and I'm raising my score to 5.

---

> > > ### Author Response · Authors · 2025-08-06
> > >
> > > Thank you for the positive feedback and insightful comments that help us strengthen the paper! We will incorporate the clarifications and additional experimental results in the final version.

---

### Official Review · Reviewer_V7jc · 2025-07-01

**Clarity:** 3
**Significance:** 3
**Originality:** 3
**Rating:** 5
**Confidence:** 3

**Summary:**

The paper introduces COMPFLOW, a method for knowledge transfer in RL between a source and a target domains with different dynamics. The core contribution is a composite flow matching technique that leverages optimal transport to estimate the dynamics gap between the two domains using Wasserstein distance, where the main idea is to train the flow for the target dynamics on top of the flow for the source dynamics (ie with noise replaced by samples from the latter). COMPFLOW then uses Wasserstein distance to perform both data filtering and guide online exploration to regions with large gap. Empirically, the method is tested on 3 locomotion tasks from D4RL and compares favorably to existing baselines.

**Questions:**

1. Please address limitations above
2. Following the discussion about the improvement of composite vs direct flow in the scarce-target-data setting after Theorem 3.1, could you provide intuition on why this is the case? If both source and target data are scarce, so the TV bound likely holds even if the two transition kernels may be very far, why the two FM models in composite flow would not "bias" each other?

**Ethical Concerns:**

["NO or VERY MINOR ethics concerns only"]

**Final Justification:**

I have increased my score after reading the authors' response and other reviews since:
- the authors addressed all concerned about writing clarity and promised to update the paper accordingly
- the authors provided additional experiments which I find quite valuable for the final version

**Limitations:**

yes

**Quality:**

3

**Strengths And Weaknesses:**

Strengths:
1. The paper tackles a relevant and well-motivated problem, as the setting in which we need to train policies on one domain with scarce data but have additional data under different dynamics frequently arises in practice (eg, in robotics)
2. The proposed method is novel and sound. I particularly like the idea of "chaining" two flow-matching models as a tool to improve predictions and ease the estimation of Wasserstein distance
3. The proposed method is backed by theoretical results that formally justify its main components: composite flow (Th. 3.1) and exploration bonus (Th. 3.3)
4. The paper is very well-written and easy to read (apart from some specific details I could not grasp, see questions below)
5. COMPFLOW performs well empirically. A good amount of ablations is provided

Weaknesses:
1. Flow-matching, while powerful, may introduce non-negligible computational/implementation overhead (eg estimating the Wasserstein distance requires multiple integrations of the ODE, which may be more inefficient than eg simpler density-based models). There is no discussion about that in the paper
2. The experiments are carried out only on 3 locomotion domains from D4RL. Adding manipulation and/or navigation tasks (eg from the same suite) would add great value to the experiments
3. A baseline that learns from scratch in the target domain is missing, which makes it hard to understand the benefits of transferring source samples in these domains
4. The impact of the exploration parameter \beta (Figure 5) seems very limited (curves mostly overlap). One wonders if that component is really important: would the performance drop significantly if exploration bonus is removed?
5. While the paper is generally well written, there are a few theoretical details which are hard to grasp:
   - Theorem 3.1: the MSE was not formally defined. Also, it is not clear under what conditions we should expect the TV bound assumption to hold
   - Not clear how the optimal coupling q^* is computed in L_tar(\phi)
   - It seems the goal of the FM models is to learn conditional distributions s' | s,a, but many places (eg the paragraph 169 - 178) give the impression that these models should actually capture the joint distributions of triples (s,a,s'). Which one is correct?
   - Theorem 3.3: what assumptions are made on the composite flow model to require that its estimated dynamics gap is below \kappa? Is training assumed "perfect", so that this model returns the exact dynamics gap and not an estimate?
6. A bunch of existing works for the problem considered in this paper use instance re-weighting rather than filter samples (see eg [1,2] and references therein). These are not discussed / compared empirically

[1] Tirinzoni, A., Sessa, A., Pirotta, M., & Restelli, M. (2018, July). Importance weighted transfer of samples in reinforcement learning. In International Conference on Machine Learning (pp. 4936-4945). PMLR.
[2] Xie, A., & Finn, C. (2022, November). Lifelong robotic reinforcement learning by retaining experiences. In Conference on Lifelong Learning Agents (pp. 838-855). PMLR.

---

> ### Author Rebuttal · Authors · 2025-07-31
>
> We thank the reviewer for their positive feedback, and for noting that the problem is well-motivated and the proposed method is both novel and theoretically sound. We also appreciate the recognition that our approach performs well empirically and is supported by a thorough ablation study. Below, we provide our responses to the questions.
>
> **Q1. Flow-matching, while powerful ... There is no discussion about that in the paper**
>
> Flow matching is indeed more complex than traditional density-based models. However, since we train a linear-path flow as defined in Eq. (1), the corresponding ODE can be efficiently solved using the simple Euler method with only 10 steps, resulting in minimal computational overhead. For instance, with a batch size of 256 for training the policy network and a Monte Carlo sample size of 30, the entire filtering process takes only 0.03 seconds on an A100 GPU. The advantages of using a linear-path flow are also highlighted in (Liu et al., 2023).
>
> In addition, recent studies—e.g., (Davtyan et al., 2025) —have proposed techniques to further accelerate inference in flow-based generative models by improving data-noise coupling. These developments offer promising directions for future research. We will elaborate on this point in the limitations section.
>
> *Liu, Xingchao, and Chengyue Gong. "Flow Straight and Fast: Learning to Generate and Transfer Data with Rectified Flow." The Eleventh International Conference on Learning Representations. 2023*
>
> *Davtyan, Aram, et al. "Faster inference of flow-based generative models via improved data-noise coupling." The Thirteenth International Conference on Learning Representations. 2025.*
>
> **Q2. The experiments are carried ... great value to the experiments.**
>
> We conduct an additional manipulation experiment on the Pen task from the Adroit suite (Rajeswaran et al., 2017), focusing on two types of distribution shifts: Kinematic shift, introduced by modifying the rotation ranges of all finger joints, and Morphological shift, created by shrinking the sizes of the fingers.
>
> The results are shown in the table below. As shown, our method achieves the highest reward on both tasks.
>
> | Method   | Kinematic | Morphology |
> |----------|------------------------|--------------------------|
> | SAC      | 43 ± 18                | 53 ± 14                  |
> | BC-SAC   | 73 ± 34                | 77 ± 25                  |
> | H2O      | 538 ± 162              | 191 ± 95                 |
> | BC_VGDF  | 115 ± 63               | 93 ± 31                  |
> | BC-PAR   | 53 ± 26                | 78 ± 19                  |
> | Ours     | **682 ± 108**          | **303 ± 67**             |
>
> *Rajeswaran, Aravind, et al. "Learning complex dexterous manipulation with deep reinforcement learning and demonstrations." arXiv preprint arXiv:1709.10087 (2017).*
>
>
> **Q3. A baseline that learns ... in these domains.**
>
> We provide the performance of SAC trained from scratch in the target domain below. As expected, its reward is significantly lower than our method.
>
> | Dataset | Task Name                   | SAC              | Ours             | Highest                       |
> |---------|-----------------------------|------------------|------------------|-------------------------------|
> | MR  | HalfCheetah (Morphology)    | 1457 ± 89        | 3119 ± 107       | Ours                          |
> | MR  | HalfCheetah (Kinematic)     | 2255 ± 197       | 5189 ± 262       | Ours                          |
> | MR  | Hopper (Morphology)         | 364 ± 82         | 355 ± 6          | SAC                           |
> | MR  | Hopper (Kinematic)          | 737 ± 547        | 1024 ± 1         | 1025 ± 0 (H2O)                |
> | MR  | Walker2D (Morphology)       | 253 ± 60         | 1094 ± 791       | Ours                          |
> | M   | HalfCheetah (Morphology)    | 1467 ± 89        | 2282 ± 287       | Ours                          |
> | M   | HalfCheetah (Kinematic)     | 2316 ± 92        | 5593 ± 44        | Ours                          |
> | M   | Hopper (Morphology)         | 396 ± 60         | 604 ± 173        | Ours                          |
> | M   | Hopper (Kinematic)          | 724 ± 535        | 1023 ± 2         | Ours                          |
> | M   | Walker2D (Morphology)       | 301 ± 177        | 886 ± 372        | Ours                          |
> | ME  | HalfCheetah (Morphology)    | 1392 ± 238       | 1485 ± 67        | Ours                          |
> | ME  | HalfCheetah (Kinematic)     | 2323 ± 97        | 5750 ± 84        | Ours                          |
> | ME | Hopper (Morphology)         | 359 ± 75         | 462 ± 89         | Ours                          |
> | ME  | Hopper (Kinematic)          | 724 ± 535       | 1022 ± 2         | 1031 ± 3 (H2O)                |
> | ME  | Walker2D (Morphology)       | 228 ± 51         | 648 ± 180        | 1103 ± 444 (H2O)              |
>
> *MR = Medium Replay, M = Medium, ME = Medium Expert.*
>
> **Q4. The impact of the ... bonus is removed?**
>
> We provide the rewards with and without the exploration bonus, along with 90% confidence intervals, in the table below. As shown, removing the exploration bonus leads to a clear performance drop in all 5 tasks.
>
> | Domain                             |  with best $\beta>0$        | $\beta=0$
> |-----------------------------------|-------------------------------|----------------------------------------------|
> | Hopper Medium (Friction)          | **295 ± 61**                  | 221 ± 3                                  |
> | Hopper Medium Replay (Friction)   | **275 ± 24**                  | 232 ± 4                                   |
> | Walker2d Medium Replay (Friction) | **579 ± 129**                 | 331 ± 5                                   |
> | HalfCheetah Medium (Morph)        | **2312 ± 326**                | 1944 ± 118                                |
> | Hopper Medium (Morph)             | **635 ± 147**                 | 423 ± 26                                 |
>
>
> **Q5.1. Theorem 3.1: the MSE ... TV bound assumption to hold.**
>
> Lemma G.1 in Appendix G contains the formulation of MSE. We will include it in the main text. Thanks for the suggestion.
>
> The TV bound assumption in Theorem 3.1 is more likely to hold when the number of target samples $N_{\rm tar}$ is small. This aligns well with our setting, where we only have limited interactions with the target domain. We will make this point clearer in the revised manuscript.
>
> **Q5.2. Not clear how ... in L_tar(\phi)**
>
> We provide a detailed description of this computation process in Algorithm 3 of Appendix C. In brief, we sample a batch of source and target domain data, then solve for the optimal transport plan using the Python POT package. This transport plan is subsequently used to resample source–target data pairs. We will clarify this process more explicitly in the main text in our revised manuscript.
>
> **Q5.3. It seems the goal of the ... Which one is correct?**
>
> The goal of the FM models is to learn the conditional distribution $p(s'|s, a)$. Lines 169–178 describe the construction of the cost matrix used for solving the mini-batch optimal coupling. As explained in lines 165–168, since flow matching is conditioned on $(s, a)$, but we do not have multiple target samples $s'$ for a fixed $(s, a)$, we include $(s, a)$ in the cost function following the approach in [17]. Nonetheless, the objective of the FM model remains learning $p(s'|s, a)$. We will clarify this distinction more clearly in the revised manuscript.
>
> **Q5.4. Theorem 3.3: what assumptions ... and not an estimate?**
>
> We do not assume perfect training. Instead, Theorem 3.3 explicitly accounts for estimation error, as reflected by the terms following $-\kappa$ in Equation (5). Our approach does not require the estimated dynamics gap to be smaller than $\kappa$. Rather, both our algorithm and experiments operate directly on the estimated—potentially larger and noisy—dynamics gaps. Theorem 3.3 provides a formal analysis of how such estimation errors influence performance.
>
> **Q6.A bunch of existing works ...These are not discussed / compared empirically.**
>
> [1] is a relatively early work that was not designed for deep RL tasks. It models transition dynamics using Gaussian processes and computes importance weights based on the probability ratio between the source and target domains. A policy is then trained using decision trees. As a result, the method is limited to simple environments, making it difficult to apply to our deep RL tasks.
>
> [2] focuses on a lifelong learning setting and trains a classifier to estimate importance weights. However, as noted in page 6 of [2], the weights are ultimately used to filter out samples rather than reweight them, due to numerical instability. Training a classifier to estimate importance weights is also used in our baseline H2O, which our method outperforms.
> We will include these references and the discussion in our revised manuscript.
>
> **Q7. Following the discussion ... "bias" each other?**
>
> When $N_{\rm tar}$ is small, it is challenging to accurately learn the transition probabilities of the target domain for both composite and direct flow models due to limited data. With such a small sample size, the observed transitions may not adequately capture the true distribution $p(s'|s, a)$. In this case, when the transition kernels between the source and target domains differ significantly, the limited samples may fail to reveal those differences.
>
> However, since the source and target domains share the same state space, the source domain data still provide information of the support of that space. This allows the composite flow to leverage more information about the state space, offering an intuitive explanation for why it tends to perform better.

---

> ### Author Response · Authors · 2025-08-06
>
> Thank you for the response. We also appreciate the insightful comments, which have helped strengthen our paper! We will incorporate the clarifications and additional experiments in the final version.

---

### Official Review · Reviewer_vMHa · 2025-07-03

**Clarity:** 3
**Significance:** 3
**Originality:** 3
**Rating:** 5
**Confidence:** 4

**Summary:**

This paper aims to maximize the sample efficiency and performance of reinforcement learning when there are shifted dynamics between the offline data collected from a source environment and the target environment being currently learned. To this end, it presents a new framework that measures the shifted dynamics between the two environments more accurately and principledly, and utilizes it to perform data filtering and active exploration strategies.

**Questions:**

See the above weaknesses.

**Ethical Concerns:**

["NO or VERY MINOR ethics concerns only"]

**Final Justification:**

Like other reviewers, I feel that this paper is well structured, having good theoretical explanation. Besides, the rebuttal is good for my answer. So I change my score as accept. Thanks for this valuable paper.

**Limitations:**

There are no negative societal impact.

**Paper Formatting Concerns:**

There are no paper formatting concerns.

**Quality:**

4

**Strengths And Weaknesses:**

Strengths: This paper proposes a novel framework called COMPFLOW for reinforcement learning problems that utilize offline data in dynamically heterogeneous environments. Here, it develops a robust measurement of dynamic gaps based on the Wasserstein distance by utilizing synthetic flow structures and optimal transport theory, and achieves state-of-the-art performance on 15 different RL benchmarks by performing data filtering and active exploration. It is a very interesting topic considering the theoretical approach and addressing dynamics gaps, and it can be very effective in the efficient utilization of replay buffers in RL.

Weaknesses: In this paper, the proposed method is very effective in reducing the dynamic gap; however, it would be beneficial to provide more evidence for the practical costs of data filtering in the replay buffer and the performance improvement due to Large Dynamics Gap Exploration Reducing the Performance Gap (Theorem 3.3), although it is discussed in limitations of this draft. It is experimentally clear that strengthening exploration in areas where the dynamic gap is large helps improve performance, but this approach has the disadvantage of introducing an additional hyperparameter. If there is any way to decide the hyperparameter, it seems that it can help improve the quality of the paper.

---

> ### Author Rebuttal · Authors · 2025-07-31
>
> We thank the reviewer for the positive feedback and for recognizing the value of our theoretical approach in addressing dynamics gaps. We appreciate the acknowledgment that this is an interesting and promising direction, particularly for improving the efficient utilization of replay buffers in reinforcement learning. We provide a detailed response for each question as below.
>
> **Q1. In this paper, the proposed method is very effective in reducing the dynamic gap; however, it would be beneficial to provide more evidence for the practical costs of data filtering in the replay buffer and the performance improvement due to Large Dynamics Gap Exploration Reducing the Performance Gap (Theorem 3.3), although it is discussed in limitations of this draft.**
>
> The practical cost of data filtering is relatively small. For example, with a batch size of 256 for training the policy network and a Monte Carlo sample size of 30, the entire filtering process takes just 0.03 seconds on an A100 GPU. We will highlight in the paper that this efficiency is due to the simplicity of our flow training objective, which follows a linear path. As a result, solving the corresponding ODE at inference time is very easy—A basic Euler method with just 10 time steps is sufficient. The advantages of using a linear-path flow are also highlighted in (Liu et al., 2023).
>
> In addition, recent studies—e.g., (Davtyan et al., 2025) —have proposed techniques to further accelerate inference in flow-based generative models by improving data-noise coupling. These developments offer promising directions for future research.
>
> Furthermore, Figure 6 in the appendix presents a hyper-parameter study of the exploration parameter $\beta$. As shown, removing the exploration bonus degrades performance consistently.
>
> *Liu, Xingchao, and Chengyue Gong. "Flow Straight and Fast: Learning to Generate and Transfer Data with Rectified Flow." The Eleventh International Conference on Learning Representations. 2023*
>
> *Davtyan, Aram, et al. "Faster inference of flow-based generative models via improved data-noise coupling." The Thirteenth International Conference on Learning Representations. 2025.*
>
> **Q2. It is experimentally clear that strengthening exploration in areas where the dynamic gap is large helps improve performance, but this approach has the disadvantage of introducing an additional hyperparameter. If there is any way to decide the hyperparameter, it seems that it can help improve the quality of the paper.**
>
>
> In the future, we could explore automatically optimizing this parameter by drawing on techniques developed for entropy temperature tuning in the SAC framework—for example, the metagradient-based approach proposed in Meta-SAC (Wang and Ni, 2020). We will include this point in the discussion of limitations in our revised manuscript.
>
> *Wang, Yufei, and Tianwei Ni. "Meta-sac: Auto-tune the entropy temperature of soft actor-critic via metagradient." arXiv preprint arXiv:2007.01932 (2020).*

---

> > ### Comment · Reviewer_vMHa · 2025-08-05
> > **Answer about the author's rebuttal**
> >
> > I appreciate the explanation against my concerns. Considering the answer and other explanations about other reviewers, I will keep my rate as "Accept." Thanks for your hard works.

---

> ### Author Response · Authors · 2025-08-06
>
> We sincerely thank the reviewer for the positive feedback and valuable insights, which will help us improve the paper! We will incorporate those clarifications in the final version.

---

### Decision · Program_Chairs · 2025-09-17

**Decision:**

Accept (spotlight)

**Comment:**

This paper represents a clear technical contribution with several distinguishing qualities:
* Methodological Innovation: The composite flow approach elegantly solves the disjoint support problem that plagues existing methods
* Theoretical Rigor: Solid theoretical foundations with practical algorithmic implications
* Empirical Strength: Consistent, significant improvements across diverse benchmarks
* Real-World Relevance: Addresses a fundamental challenge in transfer learning and domain adaptation for RL